# Mesoporous multimetallic nanospheres with exposed highly entropic alloy sites

Yunqing Kang[1,2], Ovidiu Cretu[1], Jun Kikkawa[1], Koji Kimoto ®[1], Hiroki Nara[1], Asep Sugih Nugraha[3], Hiroki Kawamoto[4], Miharu Eguchi[2], Ting Liao[5] ✉, Ziqi Sun ®[6], Toru Asahi[2] & Yusuke Yamauchi ®[2,3,7] ✉

Multimetallic alloys (MMAs) with various compositions enrich the materials library with increasing diversity and have received much attention in catalysis applications. However, precisely shaping MMAs in mesoporous nanostructures and mapping the distributions of multiple elements remain big challenge due to the different reduction kinetics of various metal precursors and the complexity of crystal growth. Here we design a one-pot wet-chemical reduction approach to synthesize core–shell motif PtPdRhRuCu mesoporous nanospheres (PtPdRhRuCu MMNs) using a diblock copolymer as the soft template. The PtPdRhRuCu MMNs feature adjustable compositions and exposed porous structures rich in highly entropic alloy sites. The formation processes of the mesoporous structures and the reduction and growth kinetics of different metal precursors of PtPdRhRuCu MMNs are revealed. The PtPdRhRuCu MMNs exhibit robust electrocatalytic hydrogen evolution reaction (HER) activities and low overpotentials of 10, 13, and 28 mV at a current density of 10 mA cm$^{-2}$ in alkaline (1.0 M KOH), acidic (0.5 M H$_2$SO$_4$), and neutral (1.0 M phosphate buffer solution (PBS)) electrolytes, respectively. The accelerated kinetics of the HER in PtPdRhRuCu MMNs are derived from multiple compositions with synergistic interactions among various metal sites and mesoporous structures with excellent mass/electron transportation characteristics.

Multimetallic alloy (MMA) nanomaterials have been widely used in various applications, including catalysis, photonics, and biomedicine, due to their excellent physicochemical properties[1,2]. The compositional flexibility and multielemental synergy of MMAs offer more opportunities for optimizing properties and overcoming the limitations of single-element metals[3,4]. The increased number of metal elements in MMA probably increases entropy. Recently, the entropy-induced cocktail effects of MMA materials have finely regulated the adsorption of reactants and the intermediates bound to active sites, thereby improving catalytic activity, selectivity, and stability[5,6]. In particular, nanostructured engineering of MMAs, such as ultrasmall nanoparticles[7], core–shell nanoparticles[8], nanowires[9], and subnanometer ribbons[10], is an effective method for improving their performance. It is necessary to design nanostructured MMA with tailored

[1]Research Center for Materials Nanoarchitectonics and Research Center for Advanced Measurement and Characterization, National Institute for Materials Science (NIMS), 1-1 Namiki, Tsukuba, Ibaraki 305-0044, Japan. [2]Faculty of Science and Engineering, Waseda University, 3-4-1 Okubo, Shinjuku, Tokyo 169-8555, Japan. [3]Australian Institute for Bioengineering and Nanotechnology (AIBN) and School of Chemical Engineering, The University of Queensland, Brisbane, QLD 4072, Australia. [4]Hitachi High-Tech Corporation, 882, Ichige, Hitachinaka-shi, Ibaraki 312-0033, Japan. [5]School of Mechanical, Medical and Process Engineering, Queensland University of Technology, Brisbane, QLD 4001, Australia. [6]School of Chemistry and Physics, Queensland University of Technology, Brisbane, QLD 4001, Australia. [7]Department of Materials Process Engineering, Graduate School of Engineering, Nagoya University, Nagoya 464–8603, Japan. ✉e-mail: t3.liao@qut.edu.au; y.yamauchi@uq.edu.au

morphologies and compositions to increase the richness of nanostructures and explore the structure−performance relationship.

Mesoporous metallic materials with high surface areas, adjustable pore structures, and efficient mass/electron transportation processes are important components in the field of nanomaterials[11,12]. Regarding catalysis, reasonable synthesis conditions facilitate obtaining mesoporous metallic alloys with different compositions and morphologies, such as core−shell PdPt[13,14], hollow PdAgCu[15], and film PtPdAu[16], to improve properties relative to monometallic components. Despite great progress, the classic mesoporous metallic alloys prepared by a wet-chemical reduction method contain mainly two or three metal elements[17–21]. Recently, as representative MMA materials, high-entropy alloys (HEAs) that contain five or more elemental components with atomic ratios ranging from 5 to 35% have received a great amount of attention because of their multielemental compositions[22,23]. However, increasing the metal elements to five or more in mesoporous MMA is a major challenge, probably because different metal ions have various physicochemical properties and standard redox potentials ($E^0$), leading to complicated co-reduction kinetics. In addition, in the multiple metal precursor system, the reduction rate of a metal species cannot be simply evaluated by $E^0$; it may be influenced by the coordination environments, surface energies, organic additives, and reducing agent selection[24,25]. The reduction process of metal species becomes more complicated in the preparation of MMA with multiple elements, leading to difficulties in controlling the nucleation and crystal growth processes. Very recently, a mesoporous noble-metal-based MMA (PtPdRhRuIr HEA) with a uniform pore size and a large surface area was successfully synthesized using polymer-templated spray-drying through following the annealing strategy by Faustini et al.[26]. Nevertheless, the synthesis of MMA mesoporous nanospheres (MNs) with multiple elements under milder synthesis conditions, such as wet-chemical reduction in the solution phase, is still rarely reported.

In this work, we demonstrate a one-pot wet-chemical reduction method for the synthesis of multimetallic PtPdRhRuCu MNs (PtPdRhRuCu MMNs) by using block copolymer micelles (poly (ethylene oxide)-block-poly(methyl methacrylate), PEO-b-PMMA) as the soft template. The obtained typical PtPdRhRuCu MMNs possess well-distributed large exposed mesopores (≈23 nm), and the pore size is tunable. As a proof of the catalysis concept, PtPdRhRuCu MMNs achieve high activity and stability levels toward the electrocatalytic hydrogen evolution reaction (HER) in all pH ranges, outperforming monometallic Pt MNs, other alloyed MMNs, and several alternative MMA-based HER electrocatalysts. The abundant exposed mesopores of PtPdRhRuCu MMNs allow efficient ion migration during the HER and the high densities of highly entropic alloy sites (HEASs), which are assumed to have lower energy barriers for water dissociation and optimal hydrogen binding energy due to multielemental synergy.

## Results and discussion

### Morphological and structural characterizations

The PtPdRhRuCu MMNs were synthesized via a simple one-pot chemical reduction method, in which PEO$_{10500}$-b-PMMA$_{18000}$ micelles and L-ascorbic acid (L-AA) were used as the pore-directing and reducing agents, respectively (Fig. 1a and Supplementary Fig. 1). Briefly, PEO$_{10500}$-b-PMMA$_{18000}$ was completely dissolved into N,N-dimethylformamide (DMF), forming a transparent solution. Then, aqueous solutions of metal precursors with the desired ratio and L-AA were added to the above solution for micellization (Supplementary Fig. 1a–c). The reaction solution was kept at 80 °C for 4 h in an oil bath (Supplementary Fig. 1d). The final PtPdRhRuCu MMNs were collected by centrifugation, and the micelles were washed with acetone/ethanol several times, as confirmed by Fourier transform infrared (FTIR) spectroscopy (Supplementary Fig. 2). Scanning electron microscopy (SEM) (Fig. 1b) and high-angle annular dark-field scanning transmission electron microscopy (HAADF−STEM) (Fig. 1c) revealed that the

synthesized PtPdRhRuCu MMNs displayed mesoporous spherical morphologies (≈128 nm) with uniformly sized mesopores (≈23 nm) (Supplementary Fig. 3). According to the small-angle X-ray scattering measurement (SAXS) (Supplementary Fig. 4), a pore-to-pore spacing of 39 nm was calculated in PtPdRhRuCu MMNs, with a single peak at $q = 0.16$ nm$^{-1}$.

The polycrystallinity characteristics of PtPdRhRuCu MMNs were confirmed by selected-area electron diffraction (SAED) patterns (inset Fig. 1c), in which the concentric diffraction rings with bright discrete diffraction spots were indexed to a face-centered cubic (fcc) structure. Figure 1d shows a high-resolution transmission electron microscopy (HRTEM) image of PtPdRhRuCu, clearly showing the crystallinity and distinctly visible lattice fringes with interplanar crystal spacings of 0.22 and 0.19 nm, corresponding to the (111) and (200) planes of fcc PtPdRhRuCu MMNs, respectively. The corresponding fast Fourier transform (FFT) patterns (selected from the outside porous wall region) (inset Fig. 1d) demonstrated the fcc structures of PtPdRhRuCu MMNs. Both atomic steps and kinks could be observed in the HRTEM image of crystallized PtPdRhRuCu MMNs by focusing on the edges of the mesopore (Supplementary Fig. 5). These stepped and kinked sites, which potentially originated from low-coordination atoms or lattice distortions, were proven to be critical factors for high-performance electrocatalysts[24,27].

The elemental mapping (Fig. 1e and Supplementary Fig. 6) revealed that Pt, Pd, Rh, Ru, and Cu in PtPdRhRuCu MMNs exhibited a core−shell structure with a Pd-rich core and Rh/Ru-rich shell; Pt and Cu were distributed throughout the nanoparticle, as confirmed by the line profile results (Fig. 1f, g). Similar elemental distributions could be observed in multiple metal nanospheres, demonstrating the consistency of the elemental distributions in PtPdRhRuCu MMNs (Supplementary Fig. 7). The formation of the core−shell structure could be attributed to the different reduction properties of each metal element during the chemical reduction process, as discussed later. The powder X-ray diffraction (XRD) pattern (Fig. 1h) shows that the PtPdRhRuCu MMNs exhibited metallic fcc structures with four diffraction peaks at $2\theta = 40.7°$, $47.2°$, $69.3°$, and $83.2°$ that were assigned to the (111), (200), (220), and (311) planes, respectively, which consistent with TEM result. No separated XRD peaks from metallic Pt, Pd, Rh, Ru, Cu, or metal oxides were observed, revealing that the PtPdRhRuCu MMNs employed a single-phase alloy structure without phase segregation.

The final atomic ratio of each element in PtPdRhRuCu MMNs was determined from inductively coupled plasma–optical emission spectrometry (ICP−OES), in which the ratio of Pt:Pd:Rh:Ru:Cu was 23:22:20:13:22, i.e., Pt$_{23}$Pd$_{22}$Rh$_{20}$Ru$_{13}$Cu$_{22}$ (Supplementary Table 1). Generally, the mixed configuration entropy ($\Delta S_{mix}$) could be expressed as follows[28]:

$$\Delta S_{mix} = -R \sum_{i=1}^{n} x_i \ln x_i \qquad (1)$$

where R is the gas constant and $x_i$ represents the molar concentration of each elemental component. According to the ICP−OES results, the $\Delta S_{mix}$ in bulk PtPdRhRuCu MMNs was larger than 1.59 R (i.e., $\Delta S_{mix} > 1.59$ R), making them HEA materials[22]. The elemental fraction determined by ICP−OES was consistent with the transmission electron microscopy–energy dispersive spectroscopy (TEM−EDS) (right side of Fig. 1f) and SEM–EDS results. Despite the gradient distributions of the multimetal elements in PtPdRhRuCu MMNs, the externally exposed mesoporous structure still ensured the accessibility of many HEASs, which will be further discussed later.

Unlike the previously reported single-crystal HEA (CrMnFeCoNi)[29], for which it was possible to obtain atomic resolution EDS maps, this process was more difficult in our MMN sample due to the presence of many different types of atoms overlapping each other. To see this gradient change in composition in our PtPdRhRuCu MMN

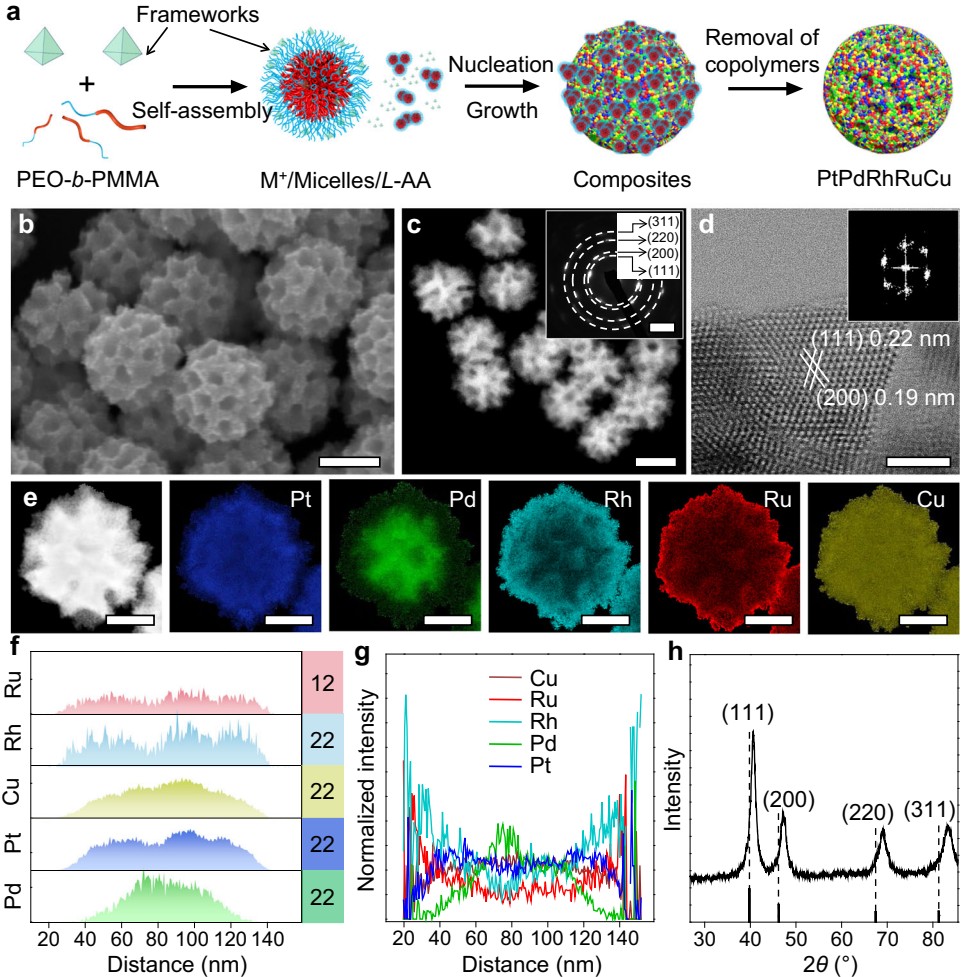

**Fig. 1 | Structural characterization of PtPdRhRuCu MMNs. a** Schematic illustration of the synthesis of PtPdRhRuCu MMNs. **b** SEM (scale bar: 100 nm), **c** HAADF–STEM (scale bar: 100 nm). Inset: The corresponding SAED pattern (scale bar: 5 nm⁻¹), **d** HRTEM (scale bar: 2 nm) (FFT patterns was shown inset), **e** atomic elemental mapping images (scale bar: 50 nm), **f** line-scanning results, **g** normalized atomic compositional profile, and **h** powder XRD pattern (compared to fcc Pt, PDF#04-0802) of the PtPdRhRuCu MMNs. Source data are provided as a Source data file.

sample clearly, we performed further analysis of one pore on the edge of one pore of the PtPdRhRuCu MMN. As shown in Fig. 2, Rh and Ru were clearly observed around the edge of the mesopore, and Pd was present deep in the mesopore; Pt and Cu were almost evenly distributed throughout the pore. This result coincided with the observed whole nanosphere in Fig. 1e–g. Importantly, the PtPdRhRuCu MMNs had core–shell nanosphere structures and large exposed mesopores, which extended from the exterior to the interior of the structures (Fig. 1 and Supplementary Fig. 3). Therefore, the PtPdRhRuCu MMNs provided abundant channels for the adsorption and diffusion of the reactants, allowing reactants to diffuse deep inside pores and nanospheres. Furthermore, the compositions of the elements tended to vary in a multivariate manner from the outside to the inside, and the elements were abundant in the interior of exposed large pores. Typically, we measured the variation in $\Delta S_{mix}$ from the outside to the inside of the single mesopore as the selected area (1, 2, 3, and 4) of Fig. 2a. The results in Fig. 2b demonstrate that $\Delta S_{mix}$ increased gradually from the outer shell to the inside, reaching a maximum value (i.e., $\Delta S_{mix} \approx 1.61$ R) in the deep site of the pore (the corresponding cartoon of one mesopore of PtPdRhRuCu MMNs shown in Fig. 2c). Continuing inward to the core region decreased the value of $\Delta S_{mix}$ because Pd dominated inside the core. The inner region of each mesopore of the PtPdRhRuCu MMN could offer many HEASs, which probably facilitated the maximization of synergistic effects (or cocktail effects) between different

metals to boost the catalytic performance. The abundant large exposed mesopores probably offered effective and separated active sites.

X-ray photoelectron spectroscopy (XPS) was used as a surface-sensitive probe to investigate the chemical state and composition of the PtPdRhRuCu MMN surface. As shown in Supplementary Fig. 8, all the metal elements in the PtPdRhRuCu MMNs showed dominant metallic states. Notably, the binding energies for the Pt⁰ 4f XPS spectra of PtPdRhRuCu MMNs showed positive shifts of 0.3 eV relative to bare mesoporous Pt (Supplementary Fig. 8a). The results demonstrated that the Pt in PtPdRhRuCu MMNs served as an electron acceptor to receive electrons from other metals, probably due to the electronic interactions among the Pt, Pd, Rh, Ru, and Cu atoms. The atomic ratio of Pt, Pd, Rh, Ru, and Cu on the surfaces of PtPdRhRuCu MMNs (determined by XPS) were 18:1.0:48:22:11, which was significantly different from the ICP–OES result (i.e., 23:22:20:13:22). Compared to other metals, the XPS signals of Pd appeared to be noisy due to the low content ratio of Pd near the surface. This phenomenon occurred because Pd precursors were first reduced during the chemical reduction process to form a Pd-rich core, which will be further discussed later. Nevertheless, certain interactions, such as the modification of the electronic structure and rearrangement of surface atoms, between the core and shell were reported to influence the catalytic performance[20,30].

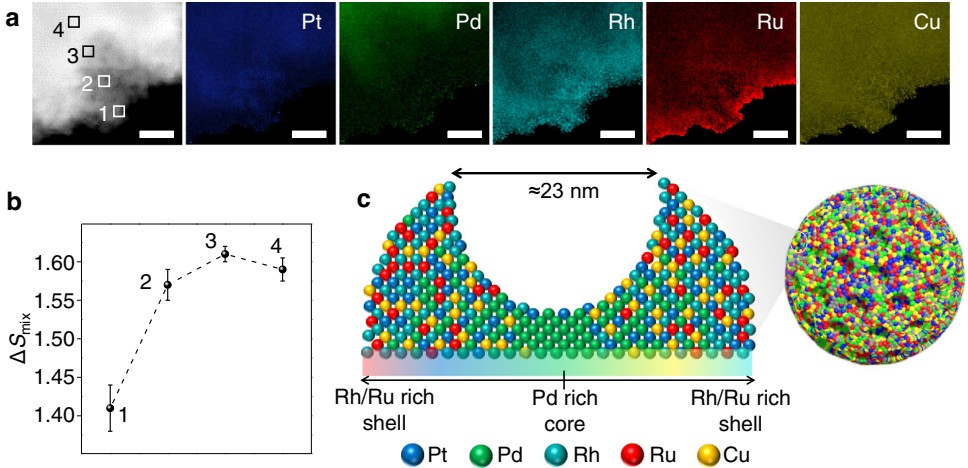

**Fig. 2 | Elemental distributions at one pore of PtPdRhRuCu MMNs.**
**a** HAADF–STEM image and EDS maps at one edge pore of a PtPdRhRuCu MMN (scale bar: 10 nm). **b** Corresponding $\Delta S_{mix}$ value at the selected area (i.e., 1, 2, 3, and 4) of (**a**). **c** Schematic of the elemental distribution around one mesopore of the PtPdRhRuCu MMN. Error bar in (**b**) correspond to the standard deviation based on the measurements at three points within the selected region. Source data are provided as a Source data file.

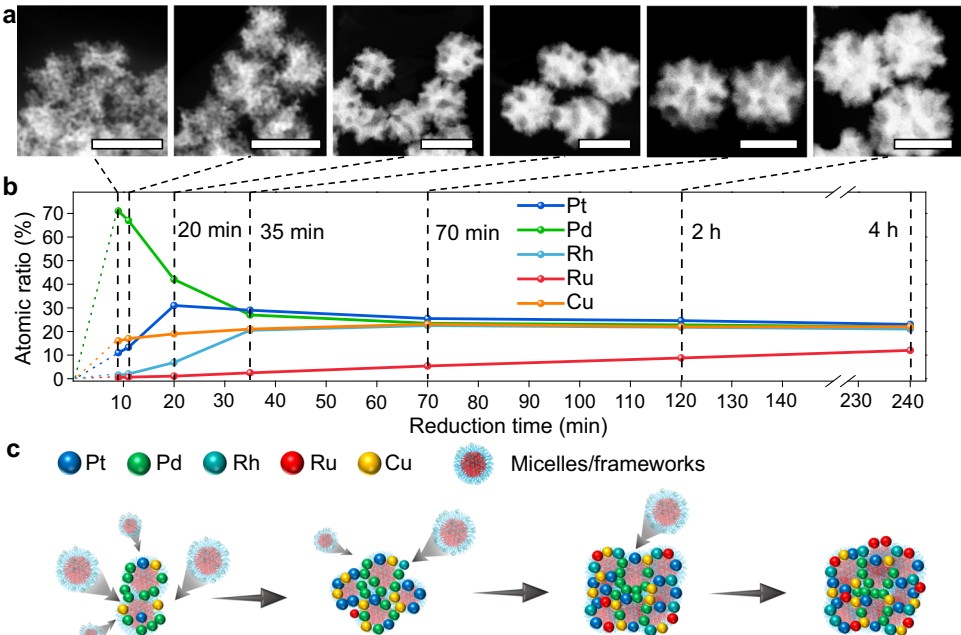

**Fig. 3 | Reduction process for PtPdRhRuCu MMNs. a** Time-dependent HAADF images (scale bars: 100 nm) and **b** composition distributions (determined by SEM– EDS) of PtPdRhRuCu MMNs. **c** Schematic illustration of the formation of PtPdRhRuCu MMNs. Source data are provided as a Source data file.

## Reduction process study

An in-depth understanding of how different metals were deposited during chemical reduction was essential for the precise design of alloyed catalysts. This process could be even more complicated in MA materials due to the multiplication of various metal precursors. In general, since different metals had different standard redox potentials ($E^0$, V vs. standard hydrogen electrode (SHE)), we could initially estimate the reduction speed of the metal ions, i.e., the more positive $E^0$ was, the easier it was to reduce the ions. However, the reduction in metal ions was controlled by $E^0$ and other aspects, such as the complicated reduction kinetics, the coordination environment, and the selection of reducing agents or surfactants[24]. For example, in the preparation of PtPd-based alloys by the wet-chemical reduction method with *L*-AA as the reducing agent[20], although Pt ($[PtCl_4]^{2-}$/Pt: +0.76 V vs. SHE) had a higher $E^0$ than Pd ($[PdCl_4]^{2-}$/Pd: +0.59 V vs. SHE)

(Supplementary Table 2), Pd ions tended to be deposited preferentially as the inner core, while Pt became the outer shell. In our reduction process, the time-dependent HAADF–TEM images in Fig. 3a show the nucleation and subsequent growth processes of the mesoporous PtPdRhRuCu sample. The time-dependent variations in the atomic fraction of metals revealed the different reduction behaviors of elements (Fig. 3b and Supplementary Fig. 9). We observed the anomalous reduction phenomenon where the Pd precursor ($[PdCl_4]^{2-}$) had a higher reduction rate than other metal precursors, including $[PtCl_4]^{2-}$, $[RhCl_6]^{3-}$, $Ru^{3+}$, and $Cu^{2+}$, leading to preformed Pd nuclei. At the initial stage (9 min), the elemental proportions of Pd, Pt, and Cu were 71, 11, and 16 at.%, respectively, while Rh (<2.0 at.%) and Ru (<1.0 at.%) were rarely observed. Interestingly, Cu was detected in large fractions (>15 at.%) at the beginning stage (9–11 min), even though its $E^0$ value ($Cu^{2+}$/Cu: 0.34 V vs. SHE) was smaller than that of Pt ($[PtCl_4]^{2-}$/Pt:

+0.76 V vs. SHE), Rh ([RhCl$_6$]$^{3-}$/Rh: +0.43 V vs. SHE), and Ru (Ru$^{3+}$/Ru: +0.39 V vs. SHE) (Fig. 3b and Supplementary Table 2). As Pd, Pt, and Cu were preferentially consumed, Rh and Ru gradually grew more on the outside of the nanosphere than on the inside without changing the lattice parameters. Specifically, Rh started to be reduced significantly after 20 min, while Ru was the slowest to be reduced. The reduction processes of these core–shell PtPdRhRuCu MMNs could be simply represented by a schematic illustration (Fig. 3c); that is, the reduction order was Pd > Pt ≈ Cu > Rh > Ru. It should be noted that Rh and Ru precursors were gradually reduced throughout the process, but were more consumed at lower concentrations of Pd, Pt, and Cu precursors, resulting in the enrichment of Rh and Ru in the shell. Overall, the above discussion revealed the specific reduction characteristics of different metals in the preparation of MMNs by a wet-chemical reduction strategy, providing guidance for the rational design of nanostructures and compositions of multimetallic alloys to achieve high-performance catalysts.

The conditions of wet-chemical reduction, including surfactants, reducing agents, organic solvents, temperature, and acids, were optimized (Supplementary Figs. 10–12). Only nonporous nanospheres could be obtained without the presence of surfactant pore-directing agents (i.e., PEO$_{10500}$-$b$-PMMA$_{18000}$, see Supplementary Information for details) (Supplementary Fig. 10a). The choice of surfactant was important. F127 caused nanospheres with decreased mesopores (Supplementary Fig. 10b), probably because of its small molecular weight, whereas polyvinylpyrrolidone formed irregular nanostructures (Supplementary Fig. 10c). Different conditions, such as reducing agents, organic solvents, and acids, affected the final morphology (Supplementary Fig. 10d–g). For example, the strong reducing agent (dimethylamine borane) yielded agglomerated nanoparticles (Supplementary Fig. 10d), and formic acid generated an irregular porous structure (Supplementary Fig. 10e). The content of Ru slightly increased with increasing reduction temperature, which had no effect on the mesoporous morphology (Supplementary Fig. 11). The appropriate amount of HCl (i.e., 6.0 M 0.4 mL) was selected in the present reduction system (Supplementary Fig. 12). The above results indicated that the establishment of a suitable chemical reduction system was crucial for the formation of well-defined MMNs.

## Library synthesis and pore size control

As mentioned above, in the preparation of porous metallic alloys by a wet-chemistry strategy based on micellar self-assembly, the complexity, and difficulty of the reduction system increased with the addition of metallic elements. Finding suitable reducing conditions to balance the complex kinetic behaviors of different metals in various metal alloys was key to the synthesis of a well-defined mesoporous morphology. Supplementary Figs. 13–16 show the SEM images and XRD patterns of other monometallic (i.e., Pt, Pd, Rh, Ru, and Cu) and metallic alloys (i.e., bimetallic Pt$M$ ($M$ = Pd, Rh, Ru, Cu), trimetallic PtRu$M$ ($M$ = Pd, Rh, Cu), and tetrametallic PtRhRuM (M = Pd, Cu)) prepared by fine tuning the chemical reduction system (Supplementary Tables 3, 4). Reaction conditions, such as the choice of precursor and polymer, type of acid, and reaction temperature, changed depending on the different composition of the Pt-based metallic alloy. The XRD results showed that the incorporation of Cu into the Pt lattice could be the main factor in the positive shift of the XRD peaks (Supplementary Fig. 14e), probably due to the smaller atomic radius of Cu than other noble metals (i.e., Pd, Rh, and Ru).

The adjustable pore sizes in mesoporous materials provided possibility for applications requiring the storage or hosting of target reactant species/molecules with different sizes. Our concept based on the self-assembly of micelles could be applied to easily control the pore sizes of PtPdRhRuCu MMNs. In a typical synthesis (Fig. 1a and Supplementary Fig. 1), the block polymer PEO$_{10500}$-$b$-PMMA$_{18000}$ was first completely dissolved in DMF solution as a unimer due to the good

solubility of both PEO and PMMA groups in DMF. Then, spherical micelles with hydrophilic PEO as the shell and hydrophobic PMMA as the core were formed after adding aqueous solutions, including HCl, metal precursors, and L-AA, due to the decreased solvation of the hydrophobic PMMA segments. The hydrophilic PEO segments were used to stabilize the metal precursor solution through interactions (e.g., hydrogen bonding and ion–dipole)[31] between the metal ions/ complexes and ethylene oxide in PEO, thereby increasing the concentration of metal precursors around the micelles. Hydrophobic PMMA blocks usually served as sacrificial agents to create pores. Therefore, the final pore sizes of PtPdRhRuCu MMNs were easily obtained by adjusting the molecular weight of the PMMA group. The small molecular weight (5500 g mol$^{-1}$) of PMMA decreased the average pore size to 8 nm, and it could be expanded to 41 nm when the large molecular weight (22,000 g mol$^{-1}$) of PMMA was used (Supplementary Fig. 17a–c). The PtPdRhRuCu samples with small and large pore sizes were denoted as PtPdRhRuCu-1 and PtPdRhRuCu-2 MMNs, respectively. The variations in the porous structures in PtPdRhRuCu-1, PtPdRhRuCu, and PtPdRhRuCu-2 MMNs were observed in the N$_2$ adsorption–desorption isotherms and corresponding pore size distributions (Supplementary Fig. 17d–f), which were consistent with the HAADF–STEM results (Fig. 1e and Supplementary Fig. 17g, h). The Brunauer–Emmett–Teller (BET) surface areas of PtPdRhRuCu-1, PtPdRhRuCu, and PtPdRhRuCu-2 MMNs were 39, 27, and 18 m$^2$ g$^{-1}$, respectively. The EDS maps in Supplementary Fig. 17g, h revealed that the element distributions of both PtPdRhRuCu-1 and PtPdRhRuCu-2 MMNs were similar to those of PtPdRhRuCu MMNs.

Our synthetic strategy could be used to synthesize other MMNs, such as PtPdCuNiCo (Supplementary Fig. 18) and PtPdCuNiCoMo (Supplementary Fig. 19). Both the PtPdCuNiCo and PtPdCuNiCoMo MMN samples had nanospherical morphologies with uniformly exposed mesopores, as revealed by SEM, HAADF–STEM, and SAXS results. The XRD patterns in Supplementary Figs. 18e and 19e confirmed the single-phase alloy structure without metal or metal oxide phase segregation in the PtPdCuNiCo and PtPdCuNiCoMo MMNs.

## Electrocatalytic HER performance

Electrocatalytic water splitting was the fastest, safest, and greenest method for producing highly pure hydrogen through the cathode HER[32]. To date, Pt-based electrocatalysts are state-of-the-art HER catalysts due to their optimum hydrogen binding energy and atomic hydrogen adsorption/desorption free energy[33]. The HER activities of Pt-based electrocatalysts were highly pH dependent. In general, the HER kinetics on Pt in alkaline media were two orders of magnitude lower than those in acidic environments[34]. This phenomenon occurred because the HER under alkaline conditions was related to multiple factors, including water dissociation, the binding strength of hydroxide, hydrogen binding energy, and H$_2$ recombination[32]. The precise designs of the structures and compositions of Pt-based materials were essential for the construction of high-performance HER electrocatalysts in wide pH ranges.

We first evaluated the HER performance of monometallic catalysts and found that Pt MNs exhibited the highest activity among Rh MNs, Pd MNs, Ru NPs, and Cu NPs (Supplementary Fig. 20a). Then, Pt-based alloys (including binary, ternary, quaternary, and quinary alloys) were screened step-by-step to determine the optimum HER catalyst in alkaline media (Supplementary Fig. 20b–d). As a result, PtPdRhRuCu MMNs showed the highest intrinsic HER activity and ion migration among the Pt-based catalysts (Supplementary Fig. 20e, f).

For a typical comparison, Pt MNs and commercial 20 wt% Pt/C were selected as the reference catalysts. The HER polarization curves were obtained at a scan rate of 5 mV s$^{-1}$ in Ar-saturated 1.0 M KOH aqueous solution. As shown in the linear sweep voltammetry (LSV) curves in Fig. 4a, PtPdRhRuCu MMNs achieved much higher HER activities than Pt MNs and Pt/C. Specifically, at a current density ($j$) of

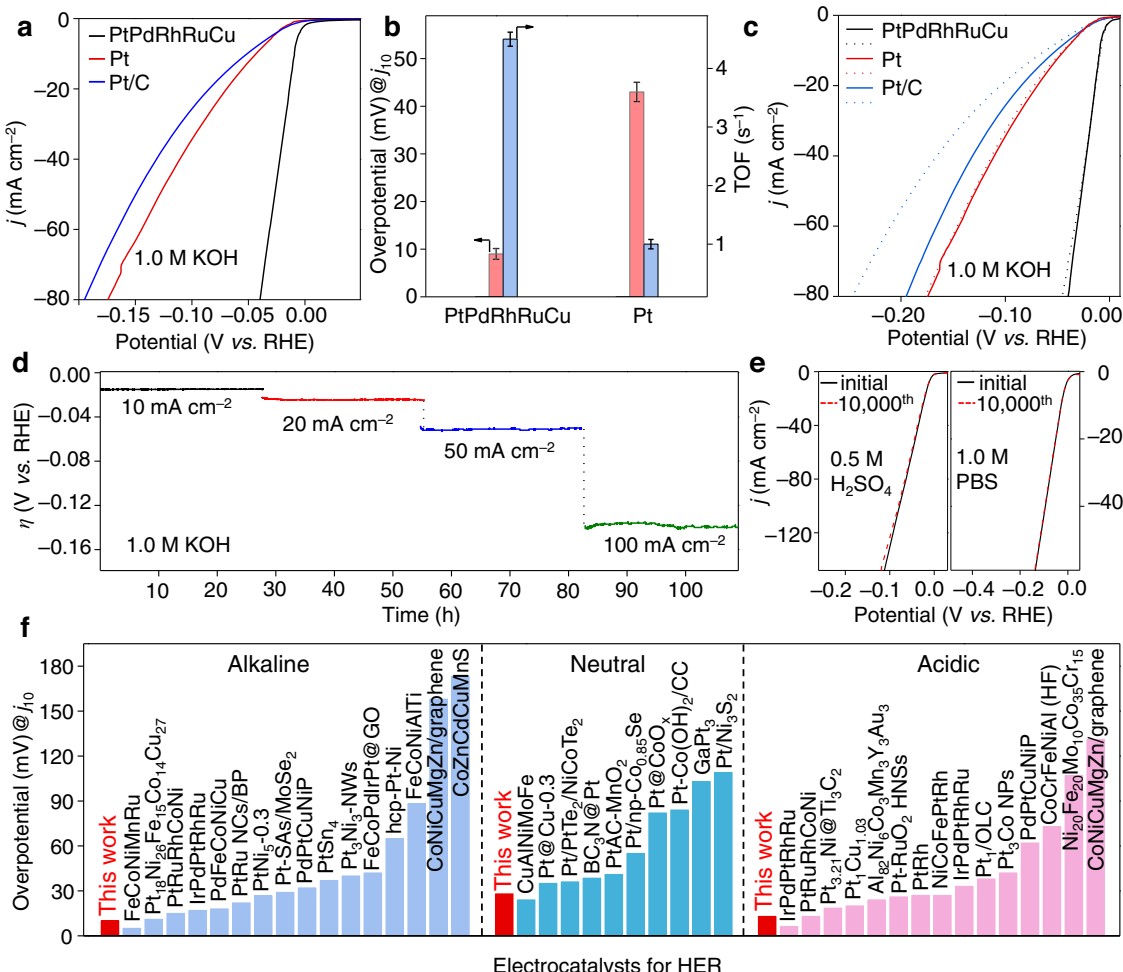

**Fig. 4 | Electrochemical HER performance. a** HER polarization curves of PtPdRhRuCu MMNs, Pt MNs, and Pt/C in 1.0 M KOH electrolyte after manual *iR* correction. **b** Comparison of overpotentials required to achieve 10 mA cm⁻² and TOF values at an overpotential of 50 mV for PtPdRhRuCu MMNs and Pt MNs. Error bars correspond to the standard deviation of three independent experiments. **c** LSV curves before (solid lines) and after 10,000 CV cycles (dashed lines) for various catalysts in alkaline media. **d** Long-term stability tests of PtPdRhRuCu MMNs

through the chronopotentiometry method at current densities from 10 to 100 mA cm⁻². **e** CV stability tests of PtPdRhRuCu MMNs in 0.5 M H₂SO₄ and 1.0 M PBS solutions. **f** Comparison of the overpotentials at 10 mA cm⁻² ($\eta_{10}$) with recently reported nanosized MMA and Pt-based HER catalysts in alkaline (1.0 M KOH), neutral (1.0 M PBS), and acidic media (0.5 M H₂SO₄), originating from Supplementary Tables 6, 7. LSV scan rate: 5 mV s⁻¹. Source data are provided as a Source data file.

10 mA cm⁻², PtPdRhRuCu MMNs had the lowest overpotential requirement of 10 mV, which was superior to the requirements of 43 and 52 mV for Pt MNs and Pt/C, respectively (Fig. 4b). For the HER in alkaline media, the hydroxide binding strength was another important descriptor, with the H binding energy. Relative to Pt, both Ru and Rh with higher oxophilicity levels had stronger abilities to cleave the HO−H bond, which facilitated the enhancement in HER activity in alkaline solution[35]. Our results showed that the coexistence of Ru and Rh significantly enhanced the alkaline HER performance (Supplementary Fig. 20b, c). Note that Ru and Rh atoms could serve as active sites instead of solely as an auxiliary catalyst through the alloying effect, as previously demonstrated for the Ru/Pt case by Zhu et al.[36]. In addition, the incorporation of smaller Cu atoms into the PtPd-based alloy caused lattice contraction, as confirmed by the positive shifts of the XRD peaks in Supplementary Figs. 14 and 21. According to the *d*-band center theory, lattice contraction would downshift the *d*-band center of Pt/Pd, weakening the adsorption energies of small intermediates (e.g., OH* and H*) and finally promoting the HER kinetics[37]. In fact, the MMA catalyst was a multielement system with an efficient synergistic effect, and the properties of single metallic elements probably changed in MMA, which was consistent with previous reports[7,38–40].

The nonporous PtPdRhRuCu nanospheres were prepared as a reference sample similar to PtPdRhRuCu MMNs without using the soft template. As shown in Supplementary Fig. 22a, the significant decline in the HER activities of nonporous PtPdRhRuCu nanospheres highlighted the effects of mesoporosity in PtPdRhRuCu MMNs, which had abundant active sites (as confirmed by enhanced electrochemical surface areas (ECSA) in Supplementary Fig. 22b–d, which was determined by the Cu underpotential deposition method)[41,42] and efficient mass/charge transport and created a rich cocktail effect in each mesopore.

The Tafel slope was analyzed to obtain more insight into the HER kinetics. As shown in Supplementary Fig. 23a, the Tafel slopes of Pt MNs were 115 mV dec⁻¹, which indicated that the Volmer−Heyrovsky mechanism operated in alkaline media[43]. In contrast, the decreased Tafel slopes in PtPdRhRuCu MMNs (87 mV dec⁻¹) indicated faster HER kinetics. The accelerated reaction kinetics of PtPdRhRuCu MMNs could be confirmed by the electrochemical impedance spectroscopy (EIS) results (Supplementary Fig. 23b), in which PtPdRhRuCu MMNs exhibited much lower charge transfer resistance values (2.7 Ω) than Pt MNs (5.8 Ω).

The intrinsic activities of PtPdRhRuCu MMNs and Pt MNs were further evaluated by turnover frequency (TOF); see Supplementary

Information for details. At an overpotential of 50 mV, PtPdRhRuCu MMNs exhibited a much higher TOF of 4.5 s$^{-1}$ than Pt MNs (1.0 s$^{-1}$) (Fig. 4b), indicating the superior intrinsic HER activities of PtPdRhRuCu MMNs. In addition, the PtPdRhRuCu MMNs exhibited an ultrahigh mass activity of 6.1 A mg$_{Pt}^{-1}$ for HER at −0.05 V vs. reversible hydrogen electrode (RHE) in an alkaline electrolyte, which was 25.4 and 32.1 times higher than that of Pt MNs and commercial Pt/C catalysts, respectively. When normalized to the masses of all noble metals (Pt+Pd+Rh+Ru in our case), PtPdRhRuCu MMNs still had a high mass activity of 2.7 A mg$_{Pt+Pd+Rh+Ru}^{-1}$ (at −0.05 V vs. RHE), which was 11.3 and 14.6 times higher than those of the Pt MNs and commercial Pt/C catalysts, respectively. As determined from N$_2$ adsorption−desorption isotherms (Supplementary Figs. 17e and 23c), the BET surface areas of PtPdRhRuCu MMNs were 27 m$^2$ g$^{-1}$, which was slightly smaller than 33 m$^2$ g$^{-1}$ for Pt MNs. The higher BET-normalized HER activities (Supplementary Fig. 23d) in PtPdRhRuCu MMNs suggested that the multielemental compositions of PtPdRhRuCu MMNs were the main factor responsible for the high electrocatalytic HER activity. Density functional theory (DFT) calculations demonstrated that the cocktail effect in multimetal elements (especially the highly entropic region inside the mesopores (Fig. 2b)) broke through the limitation of the single element to achieve low-water dissociation barriers (Volmer step) and the optimal free energy of hydrogen adsorption ($\Delta G_{H^*}$) (Heyrovsky step) value (Supplementary Fig. 24). In addition, the change in the valence electron density of each atom indicated electron transfer and redistribution upon the synergistic electronic coupling interactions in PtPdRhRuCu MMNs, as confirmed by Löwdin charge calculation, which was used to study the charge transfer effects (Supplementary Fig. 25 and Supplementary Table 5).

The size and shape of porous material may have a big influence on the catalytic properties[44]. Here, the influences of the porous structures of MMNs (i.e., PtPdRhRuCu-1, PtPdRhRuCu, and PtPdRhRuCu-2 MMNs) on HER performance were evaluated. The typical PtPdRhRuCu had higher HER activity than PtPdRhRuCu-1 MMNs with small mesopores and PtPdRhRuCu-2 MMNs with oversized mesopores (Supplementary Fig. 26a). The pore size influenced the specific surface area (Supplementary Fig. 17) and mass/electron transportation and played an important role in the density of the external exposure HEASs. As shown in Supplementary Fig. 26b, PtPdRhRuCu MMNs had a higher ECSA/EWSA (EWSA: electrochemical wettable surface area) ratio (ECSA and EWSA were proportional to the double layer capacitance values measured under dynamic and static conditions, respectively) than PtPdRhRuCu-1 and PtPdRhRuCu-2 MMNs, indicating the best ion migration in PtPdRhRuCu MMNs. Additionally, the CV scan rate dependency of the capacitance indicated that the PtPdRhRuCu MMNs maintained higher capacitance levels even under severe dynamic conditions; this finding suggested the excellent ionic paths in PtPdRhRuCu MMNs (Supplementary Fig. 26c). Although the PtPdRhRuCu-1 MMNs had higher surface areas than the PtPdRhRuCu MMNs (Supplementary Fig. 17d, e), the densities of exposed HEASs in the PtPdRhRuCu-1 MMNs were lower than those in the PtPdRhRuCu MMNs, probably due to the limited pore sizes of the PtPdRhRuCu-1 MMNs (Supplementary Fig. 27). The above results indicated that the appropriate mesopores in PtPdRhRuCu MMNs positively affected ion movement while ensuring high surface areas and the exposure of HEASs, improving HER performance.

Stability tests, including CV cycles and chronopotentiometry measurements of PtPdRhRuCu MMNs, were performed. As shown in Fig. 4c, there were no obvious shifts in the HER polarization curves of PtPdRhRuCu MMNs before and after 10,000 CV cycles. In addition, the chronopotentiometry curves operated for more than 140 h in 1.0 M KOH solution at different current densities (10–100 mA cm$^{-2}$) for PtPdRhRuCu MMNs. These results confirmed the good electrochemical durability of PtPdRhRuCu MMNs. Moreover, the mesoporous morphologies, single-phase fcc structures, metallic states, and

compositions of PtPdRhRuCu MMNs after the durability test were further characterized by TEM/HRTEM, XRD, XPS, and ICP–OES, respectively. As shown in Supplementary Figs. 28, 29 and Supplementary Table 1, the negligible changes after the HER suggested the high compositional and structural stabilities of PtPdRhRuCu MMNs for the electrocatalytic HER in alkaline media. The excellent HER stabilities of PtPdRhRuCu MMNs could be attributed to the following: (1) the exposed mesoporous structure ensured the sufficient exposure of the active sites and HEASs and inhibited their migration and agglomerative growth; (2) the stabilization effects and high metallic element diffusion barriers of MMA materials; and (3) the inhibition of the separation and etching of the metal elements by the stable crystallinity.

In addition to alkaline electrolyte, the HER performance levels of PtPdRhRuCu MMNs under acidic (0.5 M H$_2$SO$_4$) and neutral (1.0 M PBS) conditions were further investigated to evaluate its pH universality. Our PtPdRhRuCu MMNs exhibited overpotentials of 13 and 28 mV at a current density of 10 mA cm$^{-2}$ in 0.5 M H$_2$SO$_4$ and 1.0 M PBS electrolytes, respectively (Supplementary Fig. 30a–c). As shown in Fig. 4e and Supplementary Fig. 30d, e, PtPdRhRuCu MMNs displayed high CVs and long-term stabilities in both acidic and neutral solutions. In general, bare Pt catalysts had significantly higher HER performance in acid than under basic and neutral conditions[45], which was quite different from our MMN catalysts. The PtPdRhRuCu MMNs had pH-independent properties for the HER; thus, they could be applied to various types of cells for water splitting (e.g., alkaline and proton exchange membrane electrolysis). The nearly 100% Faradaic efficiencies, which were obtained by comparing the experimentally measured volume of H$_2$ with the theoretically calculated values (Supplementary Fig. 31), of the PtPdRhRuCu MMNs for pH-universal HER indicated that no side reactions occurred during electrolysis. In recent years, MMA electrocatalysts with excellent HER performance levels were demonstrated (Supplementary Table 6)[7,38–40,46–69], and materials, such as our PtPdRhRuCu MMNs, with competitive activities in alkaline, neutral, and acidic solutions have rarely been reported. Moreover, the PtPdRhRuCu MMNs still possessed excellent HER performance levels in different pH media relative to other reported Pt-based electrocatalysts (Supplementary Table 7). Figure 4f shows a comparative histogram of the HER performance levels of PtPdRhRuCu MMNs with lower overpotentials than other reported nanosized MMA and Pt-based electrocatalysts in different media, verifying that it was a highly efficient pH-universal catalyst for HER.

Based on the above discussion, the superior electrocatalytic HER performance levels of PtPdRhRuCu MMNs in the various electrolytes could be explained as follows: (1) the core–shell mesoporous structure and large exposed mesopores featured high atom utilization, rich active HEASs, and effective mass transportation; (2) the steady nanostructure and crystallinity guaranteed the long-term stability; (3) the strong synergistic electronic effect among the different atoms contributed to the fine-tuning electronic structures and influences of HER activity; and (4) the cocktail effects in multimetal elements overcame the limitation of the single element to achieve low-water dissociation barriers and the optimal $\Delta G_{H^*}$ value. This work provided a viable synthetic route for the fabrication of MMNs with turntable compositions and pore sizes. A deep understanding of the different reduction behaviors of multiple metal element precursors during chemical reduction allowed us to design high-performance nanomaterials, which could promote research on material science and catalysis.

In summary, we report a simple wet-chemical reduction route to construct core–shell PtPdRhRuCu MMNs with adjustable pore sizes and compositions using block copolymers as the soft template. We revealed the self-assembly formation process for the mesoporous morphology and the different reduction behaviors of various metals in the reduction process. The PtPdRhRuCu MMNs had uniform, well-defined, and exposed mesopores, possessing abundant HEASs,

especially on the deep side of each mesopore. The synthesized PtPdRhRuCu MMNs showed robust electrocatalytic activity and durability characteristics toward the HER in alkaline, acidic, and neutral media. Especially at 1.0 M KOH solution, the mass activities of PtPdRhRuCu MMNs reached 6.1 A mg$_{Pt}^{-1}$ at −0.05 V (vs. RHE), which was 25.4 times higher than those of Pt MNs. The catalytic performance levels of PtPdRhRuCu MMNs with multiple metals involved were dependent on a cooperative effect from the active metal site and its coordinated metals rather than the effects from separate metals, which was beneficial for pH-universal HER catalytic activity and stability. The present strategy could be extended to synthesize other MMAs, such as PtPdCuNiCo and PtPdCuNiCoMo MMNs. Our results could guide the design of functional nanostructured MMA with rational morphologies and controlled compositions by chemical reduction methods for targeted catalytic applications and beyond.

## Methods

### Chemicals
All reagents were used as purchased without further purification. Potassium tetrachloroplatinate(II) (K$_2$PtCl$_4$, 98%), sodium tetrachloropalladate(II) (Na$_2$PdCl$_4$, 98%), sodium hexachlororhodate(III) (Na$_3$RhCl$_6$), ruthenium(III) chloride hydrate (RuCl$_3$·xH$_2$O, 99.98% trace metals basis), palladium(II) chloride (PdCl$_2$, ≥99.9%), copper(II) chloride dihydrate (CuCl$_2$·2H$_2$O, ≥99.0%), nickel(II) chloride hexahydrate (NiCl$_2$·6H$_2$O, ≥97%), cobalt(II) chloride hexahydrate (CoCl$_2$·6H$_2$O, ≥97%), sodium molybdate dihydrate (Na$_2$MoO$_4$·2H$_2$O, ≥99%), L-AA (99%), dimethylamine borane (97%), hydrazine monohydrate (N$_2$H$_4$·H$_2$O, N$_2$H$_4$ 64−65%, reagent grade, 98%), and Nafion™ perfluorinated resin solution (5 wt% in mixture of lower aliphatic alcohols and water, contains 45% water) were purchased from Sigma Aldrich. The block copolymers, polystyrene-*b*-poly(ethylene oxide) (PS$_{5000}$-*b*-PEO$_{2200}$), poly(ethylene oxide)-*b*-poly(methyl methacrylate) (PEO-*b*-PMMA) with different molecular weights of PMMA blocks, including PEO$_{10000}$-*b*-PMMA$_{5500}$, PEO$_{10500}$-*b*-PMMA$_{18000}$ and PEO$_{10500}$-*b*-PMMA$_{22000}$ (the subscript numbers indicate the average molecular weight (g mol$^{-1}$) of the corresponding blocks), were obtained from Polymer Source. DMF, Tetrahydrofuran (THF), and acetone were purchased from Nacalai Tesque, Inc.

### Synthesis of multimetallic PtPdRhRuCu MMNs
The PtPdRhRuCu MMNs were synthesized by an assembly of diblock copolymer micelles via a wet-chemical approach. In typical synthesis, 5 mg of PEO$_{10500}$-*b*-PMMA$_{18000}$ was dissolved in 0.8 mL DMF under sonication in a 20 mL vial. Then, 0.4 mL HCl (6.0 M), 0.2 mL H$_2$O, and aqueous solutions of 0.4 mL K$_2$PtCl$_4$ (40 mM), 0.4 mL Na$_2$PdCl$_4$ (40 mM), 0.4 mL Na$_3$RhCl$_6$ (40 mM), 0.6 mL RuCl$_3$·xH$_2$O (40 mM), and 0.4 mL CuCl$_2$·2H$_2$O (40 mM) were added to the above solution. After gently shaking, 2 mL L-AA solution (0.1 M) was injected. The reaction was kept at 80 °C for 4 h. The synthesized products were centrifuged at 14,000 × *g* for 10 min and then washed with a mixture of acetone and water several times to remove the soft template. Nonporous PtPdRhRuCu nanospheres were prepared by using similar procedures without using the diblock copolymer.

### Synthesis of monometallic MNs and nonporous NPs
**Pt MNs.** Briefly, 5 mg of PEO$_{10500}$-*b*-PMMA$_{18000}$ was completely dissolved in 0.4 mL DMF in a 20 mL vial. Next, 0.1 mL HCl (6 M), 0.5 mL H$_2$O, 2.0 mL K$_2$PtCl$_4$ (40 mM), and 2.0 mL L-AA solution (0.1 M) were added in the aforementioned solution. This solution was kept at 55 °C for 4 h. The black products were collected by centrifugation and washed several times with an acetone/water mixture.

**Pd MNs.** Briefly, 5 mg of PS$_{5000}$-*b*-PEO$_{2200}$ was completely dissolved in 0.2 mL THF in a 20 mL vial. Then, 0.1 mL HCl (2 M), 0.7 mL H$_2$O, 2.0 mL H$_2$PdCl$_4$ (40 mM), and 2.0 mL L-AA solution (0.1 M) were added in the aforementioned solution. This solution was kept at 40 °C for 4 h. The black products were collected by centrifugation and washed several times with an acetone/water mixture.

**Rh MNs**
The Rh MNs were synthesized as followings[31]: Briefly, 5 mg of PEO$_{10500}$-*b*-PMMA$_{18000}$ was completely dissolved in 0.6 mL DMF. Next, Then, the aqueous solutions, including 0.4 ml of deionized water, 1.0 ml of Na$_3$RhCl$_6$ (40 mM), and 1.0 ml of L-AA solution (0.1 M) were added to the above DMF solution in sequence. This solution was kept at 60 °C for 12 h in a water bath. The black products were collected by centrifugation and washed several times with an acetone/water mixture.

**Ru and Cu NPs.** Since both Ru$^{3+}$ and Cu$^{2+}$ ions were not reduced by L-AA at present experimental conditions. The Ru and Cu NPs were synthesized by a chemical reduction method using N$_2$H$_4$·H$_2$O as the reducing agent. Typical for the synthesis of Ru NPs, 2.0 mL aqueous RuCl$_3$·xH$_2$O (40 mM) was added in 3.0 mL H$_2$O under stirring. Then, the 2.0 mL N$_2$H$_4$·H$_2$O was added dropwise to the above solution at room temperature. The reaction proceeded for 6 h with stirring, and the products were collected by centrifugation and washed several times with an ethanol.

The Cu NPs was prepared in the same way as Ru NPs except that the metal precursor is replaced by CuCl$_2$·2H$_2$O.

### Synthesis of bimetallic PtM (M = Pd, Rh, Ru, Cu) MNs
**PtCu MNs.** The PtCu MNs were synthesized by following steps[70]. In brief, 5 mg of PEO$_{10500}$-*b*-PMMA$_{18000}$ was completely dissolved in 0.4 mL DMF in a 20 mL vial. Next, 0.2 mL HCl (6 M), 0.4 mL H$_2$O, 1.6 mL K$_2$PtCl$_4$ (40 mM), 0.4 mL CuCl$_2$·2H$_2$O (40 mM) and 2.0 mL L-AA aqueous solution (0.1 M) were added in the aforementioned solution. The feeding molar ratio of precursors was Pt:Cu = 4:1. This solution was kept at 85 °C for 4 h.

**PtPd MNs.** In brief, 5 mg of PEO$_{10500}$-*b*-PMMA$_{18000}$ was completely dissolved in 0.8 mL DMF in a 20 mL vial. Then, 0.4 mL HCl (6 M), 1.6 mL K$_2$PtCl$_4$ (40 mM), 0.4 mL Na$_2$PdCl$_4$ (40 mM), and 2.0 mL L-AA solution (0.1 M) were added in the aforementioned solution. This solution was kept at 60 °C for 4 h.

**PtRh and PtRu MNs.** In brief, 5 mg of PEO$_{10500}$-*b*-PMMA$_{18000}$ was completely dissolved in 0.4 mL DMF in a 20 mL vial. Then, 0.7 mL HClO$_4$ (2 M), 1.6 mL K$_2$PtCl$_4$ (40 mM), 0.4 mL Na$_3$RhCl$_6$ (40 mM), and 3.0 mL L-AA solution (0.1 M) were added in the aforementioned solution. This solution was kept at 50 °C for 6 h.

The PtRu MNs were prepared same with PtRh MNs except that the RuCl$_3$·xH$_2$O was instead of Na$_3$RhCl$_6$.

### Synthesis of trimetallic PtRuM (M = Pd, Rh, Cu) MNs
**PtRhRu MMNs.** In brief, 5 mg of PEO$_{10500}$-*b*-PMMA$_{18000}$ was completely dissolved in 0.4 mL DMF in a 20 mL vial. Then, 0.7 mL HClO$_4$ (2 M), 1.2 mL K$_2$PtCl$_4$ (40 mM), 0.4 mL Na$_3$RhCl$_6$ (40 mM), 0.6 mL RuCl$_3$·xH$_2$O (40 mM), and 3.0 mL L-AA solution (0.1 M) were added in the aforementioned solution. This solution was kept at 50 °C for 6 h.

**PtRuCu MNs.** The PtRuCu MMNs were prepared as same with PtRhRu MMNs except for changing the 0.4 mL Na$_3$RhCl$_6$ (40 mM) with 0.4 mL CuCl$_2$·2H$_2$O (40 mM), and the reaction temperature to 60 °C.

**PtPdRu MMNs.** The PtPdRu MMNs were prepared as followings: 5 mg of PEO$_{10500}$-*b*-PMMA$_{18000}$ was completely dissolved in 0.8 mL DMF in a 20 mL vial. Then, 0.2 mL water, 0.4 mL HCl (6 M), 1.2 mL K$_2$PtCl$_4$ (40 mM), 0.4 mL Na$_2$PdCl$_4$ (40 mM), 0.6 mL RuCl$_3$·xH$_2$O (40 mM), and 2.0 mL L-AA solution (0.1 M) were added in the aforementioned solution. This solution was kept at 70 °C for 6 h.

## Synthesis of tetrametallic PtRhRuM (*M* = Pd, Cu)

**PtPdRhRu and PtPdRuCu MMNs.** The PtPdRhRu MMNs were prepared as followings: 5 mg of PEO$_{10500}$-*b*-PMMA$_{18000}$ was completely dissolved in 0.8 mL DMF in a 20 mL vial. Then, 0.2 mL water, 0.4 mL HCl (6 M), 0.8 mL K$_2$PtCl$_4$ (40 mM), 0.4 mL Na$_2$PdCl$_4$ (40 mM), 0.4 mL Na$_3$RhCl$_6$ (40 mM), 0.6 mL RuCl$_3$·xH$_2$O (40 mM), and 2.0 mL *L*-AA solution (0.1 M) were added in the aforementioned solution. This solution was kept at 70 °C for 6 h.

The PtPdRuCu MMNs were prepared same with PtPdRhRu MNs except that the CuCl$_2$·2H$_2$O was instead of Na$_3$RhCl$_6$ and the reaction temperature was changed to 80 °C.

## Synthesis of PtPdRhRuCu-based MMNs with adjustable pore size

The PtPdRhRuCu-based MMNs with adjustable pore size were prepared using different block copolymers. Specifically, the PtPdRhRuCu-1 MMNs were synthesized as follows: 5 mg of PEO$_{10000}$-*b*-PMMA$_{5500}$ was dissolved in 0.8 mL DMF under sonication in a 20 mL vial. Then, 0.4 mL HCl (6.0 M), 0.2 mL H$_2$O, 0.4 mL K$_2$PtCl$_4$ (40 mM), 0.4 mL Na$_2$PdCl$_4$ (40 mM), 0.4 mL Na$_3$RhCl$_6$ (40 mM), 0.6 mL RuCl$_3$·xH$_2$O (40 mM), and 0.4 mL CuCl$_2$·2H$_2$O (40 mM) were added into the above solution. After gently shaking, 2 mL *L*-AA solution (0.1 M) was injected. The reaction was then kept at 80 °C for 4 h.

PtPdRhRuCu-2 MMNs were prepared same with PtPdRhRuCu-1 MMNs except for the PEO$_{10500}$-*b*-PMMA$_{22000}$ was used to instead of PEO$_{10000}$-*b*-PMMA$_{5500}$.

## Synthesis of other MMNs

**PtPdCuNiCo MMNs.** 5 mg of PEO$_{10500}$-*b*-PMMA$_{18000}$ was completely dissolved in 0.4 mL DMF in a vial. Then, 0.2 mL water, 0.8 mL HClO$_4$ (2 M), 0.2 mL K$_2$PtCl$_4$ (40 mM), 0.2 mL Na$_2$PdCl$_4$ (40 mM), 0.2 mL CuCl$_2$·2H$_2$O (40 mM), 0.7 mL NiCl$_2$·6H$_2$O (40 mM), 0.7 mL CoCl$_2$·6H$_2$O (40 mM), and 2.0 mL *L*-AA solution (0.1 M) were added in the aforementioned solution. This solution was sealed at 110 °C for 8 h.

**PtPdCuNiCoMo MMNs.** 5 mg of PEO$_{10500}$-*b*-PMMA$_{18000}$ was completely dissolved in 0.4 mL DMF in a vial. Then, 0.2 mL water, 0.8 mL HClO$_4$ (2 M), 0.2 mL K$_2$PtCl$_4$ (40 mM), 0.2 mL Na$_2$PdCl$_4$ (40 mM), 0.2 mL CuCl$_2$·2H$_2$O (40 mM), 0.2 mL Na$_2$MoO$_4$·2H$_2$O (40 mM), 0.6 mL NiCl$_2$·6H$_2$O (40 mM), 0.6 mL CoCl$_2$·6H$_2$O (40 mM), and 2.0 mL *L*-AA solution (0.1 M) were added in the aforementioned solution. This solution was sealed at 110 °C for 8 h.

For all references sample, the polymer was removed by washing several times with an acetone/ethanol mixture. The preparation conditions of noble-metal-based MNs were listed in Supplementary Table 2.

**Characterizations.** The compositions of the PtPdRhRuCu MMN catalysts were determined by ICP–OES (Agilent 5800). SEM was conducted using a Hitachi SU-8000 microscope operating at 10 kV. SEM–EDS was observed using a flat quad EDS (5060 F, Bruker). A Smart lab X-ray diffractometer (RIGAKU) was used to collect XRD data from 20 to 90° 2θ with a step size of 0.02° at room temperature using a Cu-Kα radiation (40 kV, 30 mA) source. XPS spectra were obtained on a Thermo Scientific K-Alpha+ under an excitation source of focused monochromatic Al Kα X-ray (1486.6 eV). The calibration of binding energies was based on the C 1*s* peak energy at 284.8 eV. The pore-to-pore distance was evaluated on a SAXS (Rigaku NANO-Viewer). TEM and HRTEM observations were obtained on a JEM-ARM200F with a Cold FEG and 2 Cs correctors (CL and OL) operated at an acceleration voltage of 200 kV. TEM–EDS analysis was performed using the dual-SDD system (detection surface area 2 × 100 mm$^2$; solid angle 1.96 = 2 × 0.98). Additional TEM characterization was performed using a Thermo Fisher Scientific ThemisZ microscope, equipped with a gun monochromator, probe Cs-corrector, and dual EDS detectors with a total solid angle of 1.76. The TEM was operated in scanning (STEM)

mode at an acceleration voltage of 80 kV. In order to improve the quality of the EDS data, the probe current was increased to ≈1 nA. Nitrogen adsorption–desorption isotherms were obtained from a BELSORP-mini (BEL, Japan) at 77 K, and the pore size distributions were calculated based on the Barrett–Joyner–Halenda (BJH) model.

**Electrochemical measurements.** Electrochemical measurements (using a CHI 660EZ instrument) were conducted using a three-electrode system with a glassy carbon electrode (GCE; 3 mm in diameter) and graphite rod as the working and counter electrodes, respectively. The Hg/HgO (1.0 M KOH), saturated calomel electrode (SCE) (saturated KCl), and Ag/AgCl (saturated KCl) electrodes were used as reference electrodes in 1.0 M KOH, 0.5 M H$_2$SO$_4$, and 1.0 M PBS electrolytes, respectively. All reference electrodes were calibrated with respect to the RHE using a high-purity hydrogen-saturated three-electrode system (Supplementary Fig. 32) as follows:

$$E_{RHE} = E_{Hg/HgO} + 0.92 \text{ in } 1.0 M \text{ KOH} \tag{1}$$

$$E_{RHE} = E_{SCE} + 0.27 \text{ in } 0.5 M \text{ H}_2\text{SO}_4 \tag{2}$$

$$E_{RHE} = E_{Ag/AgCl} + 0.61 \text{ in } 1.0 M \text{ PBS} \tag{3}$$

The mesoporous nanospheres were loaded on Vulcan XC-72 carbon (mass loading of 28 wt% for typical PtPdRhRuCu MMNs, determined by ICP–OES) in a 10 mL mixture of acetone and ethanol and stirred for 8 h to ensure the uniform distribution of catalysts on the GCE. Considering that the pure mesoporous metal nanospheres tended to agglomerate in the solution and were poorly dispersed on the GCE, the use of conductive Vulcan XC-72 carbon (without HER activity) ensured a uniform dispersion of electrocatalysis on the GCE. The products were collected by centrifugation and washed with ethanol three times. After drying, each carbon-supported catalyst (5 mg) was dispersed into a solution containing a mixture, 840 μL of ethanol, 140 μL of water and 20 μL of 5 wt% Nafion, and the suspension was ultrasonicated for 1 h to generate a well-suspended ink. Then, 3.2 μL of the ink was dropped on the polished GCE and dried naturally in air. After the first activation of the catalyst by cyclic voltammetry scanning (0 to 1.2 V vs. RHE, 20 cycles), LSV was conducted at a scan rate of 5 mV s$^{-1}$ in Ar-saturated electrolytes. Tafel plots were derived from the overpotential at different ranges versus the logarithmic current density according to the corresponding LSV curves, and the Tafel slope was calculated with the following equation:

$$\eta = a + b \log|j| \tag{4}$$

where η is the overpotential, *a* is the exchange current density, *b* is the Tafel slope, and *j* is the current density. The LSV after 10,000 cycles of CV was measured to further evaluate the stability of the catalyst. EIS was performed over a frequency range of 100 k to 0.01 Hz. The resistance of the solution resulted from the Nyquist plot was used to correct the ohmic drop (*iR* compensation) for the HER measurement. For the long-term stability test, the same catalysts (1.0 mg cm$^{-2}$) were loaded on carbon fiber paper and cycled two hundred times at a scan rate of 100 mV s$^{-1}$ in different electrolytes to form stable catalysts. Then, stability tests were performed using the chronopotentiometry method without *iR* compensation at a constant current density.

The reaction product of H$_2$ was analyzed by gas chromatography (GC, Shimadzu Tracera, BID-2010) with He as the carrier gas. The far-adaic efficiency of the HER was investigated by comparing the experimental value with the theoretical calculation value of the produced H$_2$, which was measured by a water displacement method. The fixed potentials of −2.0, −2.0, and −3.0 V was applied in 1.0 M KOH, 0.5 M H$_2$SO$_4$, and 1.0 M PBS, respectively, to generate H$_2$ form water

splitting. The plot was converted into mL vs. time format using Faraday's law of electrolysis:

$$V(H_2) = QRT/(ZPF) \qquad (5)$$

where $n$ is the number of moles, $Q$ is the charge passed during electrolysis, R is 8.3145 J K$^{-1}$ mol$^{-1}$, $T$ is 296 K, $P$ is 101.325 kPa, F is Faraday's constant (96,485 C mol$^{-1}$), and $Z$ is 2 (two electrons are required to form one $H_2$ molecule).

The ECSAs of the catalysts were obtained by the Cu underpotential deposition (Cu UPD) method as follows[41]:

$$ECSA = Q_{Cu}/(m \times 420 \, \mu C \, cm^{-2}) \qquad (6)$$

where we assume a value of 420 $\mu$C cm$^{-2}$ for saturated Cu$_{upd}$ monolayer formation on active metal sites and $m$ is the metal mass loading of the catalysts on the electrode.

The number of active sites ($n$) of catalysts was determined by the Cu UPD stripping charge ($Q_{Cu}$, Cu$_{upd}$ → Cu$^{2+}$ + 2e$^-$) using the following equation:

$$n = Q_{Cu}/(2Fm) \qquad (7)$$

Once the number of active sites was obtained, the TOF (s$^{-1}$) could be calculated according to the following equation:

$$TOF(s^{-1}) = jS/(ZFnm) \qquad (8)$$

where $j$ is the current density obtained from the LSV measurement, $S$ is the surface area of the electrode (i.e., 0.0707 cm$^2$), and $Z = 2$.

**Computational details.** Spin-polarized DFT calculations were performed throughout this work as implemented in the Quantum-Espresso package[71]. The electron-ion and exchange-correlation interactions were described by ultrasoft pseudopotentials and the generalized gradient approximation with Perdew–Burke–Ernzerhof functional, respectively[72,73]. The Kohn-Sham orbitals and the charge density were represented using basis sets consisting of plane waves up to a maximal kinetic energy of 50 Ry and 400 Ry, respectively. The MMA supercell model derived from fcc phase, which consist of 100 atoms, was built up with the stoichiometry of Pt$_{23}$Pd$_{22}$Rh$_{20}$Ru$_{13}$Cu$_{22}$ as reported from the experiment. The vacuum spacing is 15 Å along the c direction to avoid the fake mirror interactions. The van der Waals correction in Grimme's DFT-D3 scheme was used to include long-range dispersion effect[74]. Gamma point was used to perform the integration in the Brillouin zone for geometric optimization[75]. All structures are optimized with convergence criteria of $1 \times 10^{-7}$ eV for the energy and $1 \times 10^{-4}$ eV Å$^{-1}$ for the force. The Gibbs free energy ($\Delta G$) of the adsorbed intermediates in Volmer–Heyrovsky mechanism is calculated by

$$\Delta G = \Delta E + \Delta E_{ZPE} - T\Delta S \qquad (9)$$

Where $\Delta E$ is the adsorption energy of the intermediate on the studied MMA metallic sites, and $\Delta E_{ZPE}$ is the zero-point energy correction for intermediate adsorption. $\Delta S$ is the vibrational entropy of adsorbed intermediates which can be derived from frequency calculations.

It should be noted that although we used the most commonly proposed mechanism with a well-defined and accepted model, the computational model become more difficult to build in complex multimetallic alloys, and the inclusion of additional factors in the model may influence the computational results.

**Reporting summary**
Further information on research design is available in the Nature Portfolio Reporting Summary linked to this article.

## Data availability
The data that support the findings of this study are available from the corresponding author upon request. Source data are provided with this paper.

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

## Acknowledgements

This work was supported by the JST-ERATO Yamauchi Materials Space-Tectonics Project (JPMJER2003). This work used the Queensland node of the NCRIS-enabled Australian National Fabrication Facility (ANFF). Ting Liao acknowledges funding support from the Australian Research Council (DP200103568 and DP230101625). This research was undertaken with the assistance of resources from the National Computational Infrastructure (NCI), which is supported by the Australian Government under the NCRIS program.

## Author contributions

Y.Y. and T.L. conceived and supervised the research. Y.Y., Y.K., and T.L. wrote the paper. Y.Y., Y.K., and T.L. designed the experiments. Y.K. and Y.Y. performed most of the experiments and data analysis. Y.K. and T.L. performed the theoretical calculations. Y.Y., Y.K., O.C., J.K., K.K., A.S.N., H.K., M.E., Z.S., H.N., and T.A. participated in experiments and discussions. All authors discussed the results and commented on the manuscript.

## Competing interests

The authors declare no competing interests.
