## [Peer Review File · Nature Communications]

Reviewer comments, first round -

Reviewer #1 (Remarks to the Author):

The authors of this contribution report a wet-chemical reduction to synthesize core-shell particles constituted of a mixture of up to five different metals, having mesopores due to the use of block copolymers as templates. The so-obtained particles, named HEA-PtPdRhRuCu MNs, are then used as catalysts for electrochemical hydrogen evolution and it is found that they are as good or even better than normal platinum, which is the benchmark metal for this topic.

I found the results quite impressive, first of all with respect to the structural control of the particle composition and morphology that the authors seem to have (see Figure 1), and most importantly concerning the measured electrocatalytic performance in terms of overpotential and achieved current densities. However, in order to become a real landmark in the field I think that manuscript needs some improvements:

-a system containing up to five different metals with changing ratios is very complex and throughout the paper it is not clear that such a complexity is really needed and why exactly the authors have chosen the reported combinations. It looks a little bit like black art, or alchemy, and in order to explain the final choice I think it would be necessary to perform additional studies of systems with a lower degree of complexity by dropping systematically one metal and keeping the rest, then another metal and keeping the rest etc.... It is difficult to believe that every metal out of the five has really a distinct and predictable role to play in the catalytic process. A multidimensional matrix of metal combinations would be really convincing, or at least a conveniently chosen subset of such a matrix could already help to rationalize the approach.

-taking into account the complexity of the system, with various ratios of metals and an inhomogeneous distribution throughout the particle volume, I don't believe that DFT calculations can really help understanding the performance of the catalyst particles. I would recommend removing this part from the manuscript because it is completely unrealistic to simulate such a system in a reliable way, and one only can expect to come up in the end with rather trivial and intuitive statements that don't need sophisticated calculations (see the sentence "The catalytic performance of HEA-Pt₂₃Pd₂₂Rh₂₀Ru₁₃Cu₂₂ MNs with multi-metal involved is not simply dependent on the each separate metal, but on a cooperative effect from the active metal site and its coordinated metals").

-The paper would gain a lot from proof reading by a native speaker, because it is full of grammatical, syntax and typing errors.

Reviewer #2 (Remarks to the Author):

In this contribution, Yamauchi et al. studied a new type of mesoporous high-entropy alloy (HEA) nanostructures for efficient pH-universal hydrogen evolution reaction (HER) electrocatalysis. The authors investigated in a very thorough and comprehensive manner the reduction and growth kinetics during the wet-chemical reduction synthesis of the mesoporous HEA. The HEA was found to exhibit excellent HER activity (and stability), which was also researched in a systematic way combining both experimental and theoretical insights. Overall, this work has high novelty in the emerging research area of HEAs (including controlled synthesis and tailored application). The results can have implications for the future development of functional HEAs with controlled nanostructures. This reviewer would like to support the publication of the manuscript, pending the below minor comments being addressed.

1. Currently the pore size of the mesoporous HEA was only supported by TEM data. While the authors conducted N₂ adsorption-desorption isotherms to analyze the BET surface area, they did not perform pore size analysis. Could the authors use the N₂ adsorption-desorption to study the pore size distribution?

2. There may be a minor inconsistency in the atomic ratios. The ICP-OES and TEM-EDS results suggest that Ru takes the lowest atomic ratio of about 12-13%, while the SEM-EDS showed that Cu takes the lowest atomic ratio (Figure S7h).

3. The atomic ratio obtained from ICP-OES was not consistently given, "23:22:20:13:22," in line

149 and "21:23:21:13:22" in line 200.

4. Compared to other elements, the XPS data of Pd appeared to be quite noisy. Is it because of the low amount of Pd near the surface? The authors might wish to add some more discussion.

5. Figure 2b, please detail how the mixed configuration entropy (ΔS_{mix}) for each point was calculated, and how the error bar was obtained.

6. Some technical terms should be explained briefly so that general readers can understand, such as "q" in line 116 and "Lowdin charge" in line 204.

7. Table S3, please cite the source for the standard redox potentials of different metals (if applicable).

8. Some figures were not mentioned or incorrectly cited in the main text. For instance, (1) Figure 2c was not cited in the text; (2) Line 155, "right site in Figure 1g". Here, the authors may have wanted to refer to Figure 1f; (3) Line 192, "Figure 2a-e" should be revised into "Figure 3a-e". (4) Line 275, "Figure S13d" should be revised into "Figure S13f"; (5) Figure S9 and S10, the effects of temperature and the amount of HCl were not discussed in the main text; (6) Line 331, "Figure S14" was incorrectly referred to.

9. Some minor writing issues (grammar and typos) need to be corrected. (1) Line 231, "71, 11%, and 16 at%" should be revised into "71, 11, and 16 at%"; (2) Line 257, "since the well solubility of PEO and PMMA segments in DMF"; (3) Line 325, "astronger"; (4) Figure 5 caption, "chronoamperometry method" should be revised into "chronopotentiometry method"; (5) Figure S18c and S20a, y axis should be checked; (6) Line 411 and line 418, "absorbed" should be revised into "adsorbed". (7) Line 417, "absorption" should be revised into "adsorption".

10. Simply out of curiosity, would it be suitable to call the reduction process "deposition"? The authors mentioned several times the terms such as deposition properties and deposition processes.

Reviewer #3 (Remarks to the Author):

The authors report the formation of nanostructured, mesoporous high entropy alloys and their application as catalysts for hydrogen evolution. The complex nature of HEAs is not lost on the authors and clearly, great lengths have been taken to characterize and optimise these materials. It would be remiss not to acknowledge that despite the high quality of the research work, there needs to be a careful examination of the grammar and sentence structure prior to acceptance. Some examples that need attention are included below.

From the results provided the mesoporous nature of the alloy clearly influences the activity for the hydrogen evolution reaction. Handily, the work compares many catalysts, their compositions and structures within the supplementary information under basic and acidic conditions and where possible more neutral pHs. One aspect of the work that wasn't discussed was how these HEAs under review were loaded onto the carbon support for some measurements. Was this strategy successful in terms of reduced leaching or nanoparticle stability. Does the conductivity of the carbon influence the measurements?

On reflection, the identification of atomic step/kink sites (Fig. S5) appears reasonable to assist in explaining the high activity. Whilst this may be a contributing factor, how confident are the authors that this is retained over the testing periods used? Rearrangement of surfaces as seen in Fig. S5 is not uncommon when catalysts are under testing conditions. The long-term testing data (Fig. S22) would suggest that this is potentially not the only factor due to the stability shown. However, Figure S21 does not have the same scope to address this. Local surface changes on the atomic scale may not be the strongest factor but the greater bulk or intermediary depth structure such as having a greater density of different metals about the pore structure do appear to matter as Figure S19 demonstrates.

The manuscript offers a promising synthetic route to mesostructured HEAs and the activity achievable for HER under various conditions is valuable. I recommend acceptance following minor corrections.

Minor issues:

86 Until now, it is still a grand challenge to precise design of HEA...rewrite needed.

166 resulting in an overlap each-other.

194 dominant metallic states along with slightly oxidation states.

249 but with a much low Ru content

Reviewer #4 (Remarks to the Author):

The article entitled "Mesoporous High-Entropy Alloy" by Kang et al. reports about mesoporous HEAs for application in catalysis. Although the work lacks the spark of significant novelty, it is timely and interesting. Few points should be addressed to enable publication:

There are several simple/grammatical errors which should be corrected during the revision process (e.g., absorbed vs adsorbed etc.).

The authors should avoid using terms like "impressively", "remarkable" etc.

The XPS analysis is less convincing. Some of the peak area ratios seem not to be correct (see e.g., Fig. 3d for Ru⁴⁺). This will directly affect the quantitative results.

Is there an optimum in pore size/BET surface area regarding the catalytic activity? What kind of pore structure is more advantageous in this context (regular vs irregular)?

Statistics should be reported for the electrochemical measurements.

Have the authors considered using a DOI approach to tailor the composition for maximum activity?

Reviewer #5 (Remarks to the Author):

The work authored by Kang and co-workers describes the synthesis of multimetallic mesoporous particles made by direct wet chemical reduction and by using diblock copolymer micelles as template. The structure and the formation mechanism was investigated. The particles exhibit excellent performances as HER electrocatalysts.

The topic is of interest for the community working on porous materials and electrocatalysis. That being said, I believe that the work can't be published at this stage since (i) the main claim is not supported by the experimental evidences, (ii) lack of originality.

Starting from the title, the main claim is the formation of HEA mesoporous materials that is defined core-shell. This is misleading since the mesoporous particles are indeed multimetallic but with composition gradients from the outer to the inner part in which the conditions for HEA are probably met only in a small part of the material. This is even more problematic when discussing about catalytic activity (discussed later on) For this reason, the authors should provide evidences that homogenous mesoporous HEA can be made. In alternative, the claim need to be tone down. The title and the all discussion must be changed by removing the claim of HEA.

In general, the materials are not homogenous and the description of structure/compositions of the materials is confusing. For instance, it is still unclear if the porous walls are polycrystalline (as mentioned in page 5 line 117) or if the full porous particle is made of a unique porous monocrystal (figure 2c).

On the novelty: polymer templated HEA mesoporous materials with homogenous composition have been recently reported in article entitled High-Entropy-Alloy Nanocrystal Based Macro- and Mesoporous Materials ACS Nano 2022, 16, 10, 15837–15849 referenced by the author in their manuscript (ref 31). The justifications used to "undermine" this recent work (not compatible with Cu and particle size due to annealing) are inaccurate. The approach proposed by Kang and co-workers still present some merits as respect to the previous work in term of synthetic conditions (in solution, milder conditions) and this can be highlighted in the introduction. However, the very same approach was reported by some of the authors in many articles for fabrication of mesoporous particles made of metals (mesoporous Rh Nat Comm 2017, mesoporous Pd Chem. Sci. 2019, 10, 4054), and several combinations of alloys (PdCu ACS Appl. Mater. Interfaces 2019, 11, 40, 36544–36552, RhCu Chem. Mater. 2018, 30, 2, 428–435, PtCu

<https://doi.org/10.1002/chem.201804305> or even Pt Pd Rh
<https://doi.org/10.1246/bcsj.20190316>). This work is a straightforward implementation of the same concept for multi metallic materials (and this can be ok) but with a lack of control and homogeneity.

The "deposition" mechanism need to be clarified. It is shown that Ru is the slowest to be deposited. In this case the author should discuss how Ru can be inserted in the fcc lattice together with the other metals (considering that Ru that crystallize in different crystal lattice). In page 12 line 261 it is indicated that the metallic ion are already complexed by the PEO chain increasing the concentration around the micelle. If the precursors are already close to the micelles, this is in contradiction with the deposition mechanism shown in figure 4. In addition scheme in figure 4 c illustrate the formation of a single nanoparticle not of a porous material. The scheme can be modified to improve the clarity for the reader.

On the electrochemical activity. DFT calculations have been performed to understand the high activity in alkaline media. However, the used model surface is based on a composition (HEA) that do not reflect the real surface of the catalyst that is not homogenous in composition. The multimetallic particles are indeed very active but the global activity is the results of the contribution of an infinite of combination of sites (gradient in composition) that can't be oversimplified as in figure 6. Here again, without providing mesoporous HEA with homogenous composition this discussion is not accurate and should be removed.

The method was extended to other metals (Ni, Co, Mo) but the here again the term HEA can't be used without providing evidences (chemical mapping at high resolution, XPS, XRD...)

In addition the following typos can be corrected:

page 9 line 192 Figure 3
page 4 line 93 tunable
page 19 line448 tunable

Dear Reviewers,

Thank you for taking the time to review our manuscript and providing your valuable feedback. We appreciate your thoughtful comments and suggestions for improvement.

We have carefully considered your feedback and have made the revisions to address the issues you raised in the Manuscript (MS) and Supplementary Information (SI) point-by-point. Please see the following response.

We hope that the revised manuscript meets your expectations and look forward to hearing from you.

Sincerely,

Yusuke Yamauchi (*Highly Cited Researcher in Materials and Chemistry*)

Professor / The University of Queensland, Australia

https://scholar.google.com.au/citations?hl=en&user=568w7h8AAAAJ&view_op=list_works

Associate Editor / J. Mater. Chem. A, RSC

Co-Editors / Chem. Eng. J. Elsevier

Honorary Group Leader / National Institute for Materials Science (NIMS), Japan

Reviewer #1:

Comment 1-1: A system containing up to five different metals with changing ratios is very complex and throughout the paper it is not clear that such a complexity is really needed and why exactly the authors have chosen the reported combinations. It looks a little bit like black art, or alchemy, and in order to explain the final choice I think it would be necessary to perform additional studies of systems with a lower degree of complexity by dropping systematically one metal and keeping the rest, then another metal and keeping the rest etc...It is difficult to believe that every metal out of the five has really a distinct and predictable role to play in the catalytic process. A multidimensional matrix of metal combinations would be really convincing, or at least a conveniently chosen subset of such a matrix could already help to rationalize the approach.

Reply:

Thank you for your comments on our paper. Your constructive criticism has helped us to improve the quality of our research. We agree that trial and error is a common approach in scientific research, and it was particularly useful in our study of multimetallic alloys. Due to the complexity composition of the PtPdRhRuCu, we found it difficult to accurately predict the role of each metal using existing theoretical models.

To overcome this challenge, we compared the activity of our multimetallic metals with other metal combinations to develop an experimental screening method. We started with a single metal material and found that metallic Pt was the best HER catalyst. Then, we gradually optimized the Pt-based alloy composition through the screening of binary, ternary, quaternary, and quinary alloys, similar to the process of alchemy that you mentioned. Eventually, we identified the optimal proportion of PtPdRhRuCu and its advantages relative to other low-complexity alloys. We hope that our approach and findings were clear to you.

Please see the modification details in **Figures S13–S16, S20, and Tables S3 and S4** in the **SI**, as well as the text in the **MS** as:

[Page 11 Line 12–24] “As mentioned above, in the preparation of porous metallic alloys by a wet-chemistry strategy based on micellar self-assembly, the complexity and difficulty of the reduction system increased with the addition of metallic elements. Finding suitable reducing conditions to

balance the complex kinetic behaviors of different metals in various metal alloys was key to the synthesis of a well-defined mesoporous morphology. **Figures S13–S16** show the SEM images and XRD patterns of other monometallic (i.e., Pt, Pd, Rh, Ru, and Cu) and metallic alloys (i.e., bimetallic PtM (M = Pd, Rh, Ru, Cu), trimetallic PtRuM (M = Pd, Rh, Cu), and tetrametallic PtRhRuM (M = Pd, Cu)) prepared by fine tuning the chemical reduction system (see preparation details in Supplementary Information and **Tables S3 and S4**). Reaction conditions, such as the choice of precursor and polymer, type of acid, and reaction temperature, changed depending on the different composition of the Pt-based metallic alloy. The XRD results showed that the incorporation of Cu into the Pt lattice could be the main factor in the positive shift of the XRD peaks (**Figure S14e**), probably due to the smaller atomic radius of Cu than other noble metals (i.e., Pd, Rh, and Ru).”

[Page 14 Line 8–13] “We first evaluated the HER performance of monometallic catalysts and found that Pt MNs exhibited the highest activity among Rh MNs, Pd MNs, Ru NPs, and Cu NPs (**Figure S20a**). Then, Pt-based alloys (including binary, ternary, quaternary, and quinary alloys) were screened step-by-step to determine the optimum HER catalyst in alkaline media (**Figure S20b–d**). As a result, PtPdRhRuCu MMNs showed the highest intrinsic HER activity and ion migration among the Pt-based catalysts (**Figure S20e, f**).”

Comment 1-2: Taking into account the complexity of the system, with various ratios of metals and an inhomogeneous distribution throughout the particle volume, I don’t believe that DFT calculations can really help understanding the performance of the catalyst particles. I would recommend removing this part from the manuscript because it is completely unrealistic to simulate such a system in a reliable way, and one only can expect to come in the end with rather trivial and intuitive statements that don’t need sophisticated calculations (see the sentence “The catalytic performance of HEA-Pt₂₃Pd₂₂Rh₂₀Ru₁₃Cu₂₂ MNs with multi-metal involved is not simply dependent on the each separate metal, but on a cooperative effect from the active metal site and its coordinated metals”).

Reply:

Thank you for your careful evaluation of DFT part. We agree with your comment that DFT calculations are typically based on classical models of single-crystal metal, and that their application

to complex multi-metal systems is extremely complex and limited. Theoretical calculations are often difficult to simulate the real structure, interface, and reaction environment. Therefore, we have decided to remove the DFT calculation results from the main text and instead provide a brief discussion. Additionally, we have provided further explanations in the **SI** to clarify this issue as: “It should be noted that although we used the most commonly proposed mechanism with a well-defined and accepted model, the computational model become more difficult to build in complex multimetallic alloys, and the inclusion of additional factors in the model may influence the computational results.”

We acknowledge that experimental results may have greater persuasive power than computational results, and we appreciate your suggestion to focus on experimental findings.

Comment 1-3: The paper would gain a lot from proof reading by a native speaker, because it is full of grammatical, syntax and typing errors.

Reply:

Thank you very much for the suggestion, we have double-checked the entire main text and fixed all the errors that we found. For enhancing the readability and professionalism of the text, we submitted the manuscript to the office of Nature Research Editing Service of Springer Nature Author Service for English editing.

Reviewer #2:

Comment 2-1: Currently the pore size of the mesoporous HEA was only supported by TEM data. While the authors conducted N₂ adsorption–desorption isotherms to analyze the BET surface area, they did not perform pore size analysis. Could the authors use the N₂ adsorption–desorption to study the pore size distribution?

Reply:

We appreciate your time and effort in carefully reviewing our submission and providing us with constructive criticism. We previously support SEM, TEM, and N₂ adsorption–desorption isotherms to investigate the pore size of typical multimetallic PtPdRhRuCu MNs (denoted as PtPdRhRuCu MMNs). Considering that the pore size of PtPdRhRuCu-based MMNs not only influence the BET surface area,

but also effect the mass/electron transportation, we further supplied the SEM, TEM, and N₂ adsorption–desorption isotherms of PtPdRhRuCu-1 (small mesopores), PtPdRhRuCu (typical middle mesopores), and PtPdRhRuCu-1 (large mesopores). The effect of pore size on the HER performance was also provided.

Please see the data in **Figures S3, S7, S17, and S26** in the **SI**, and the text in the **MS** as following:

[Page 12 Line 7–17] “The small molecular weight (5500) of PMMA decreased the average pore size to 8 nm, and it could be expanded to 41 nm when the large molecular weight (22000) of PMMA was used (**Figure S17a–c**). The PtPdRhRuCu samples with small and large pore sizes were denoted as PtPdRhRuCu-1 and PtPdRhRuCu-2 MMNs, respectively. The variations in the porous structures in PtPdRhRuCu-1, PtPdRhRuCu (typical), and PtPdRhRuCu-2 MMNs were observed in the N₂ adsorption–desorption isotherms and corresponding pore size distributions (**Figure S17d–f**), which were consistent with the HAADF–STEM results (**Figure 1e and S17g, h**). The Brunauer–Emmett–Teller (BET) surface areas of PtPdRhRuCu-1, PtPdRhRuCu (typical), and PtPdRhRuCu-2 MMNs were 39, 27, and 18 m² g⁻¹, respectively. The EDS maps in **Figure S17g, h** revealed that the element distributions of both PtPdRhRuCu-1 and PtPdRhRuCu-2 MMNs were similar to those of typical PtPdRhRuCu MMNs.”

[Page 16 Line 8–24] “The influences of the porous structures of MMNs (i.e., PtPdRhRuCu-1, PtPdRhRuCu, and PtPdRhRuCu-2 MMNs) on HER performance were evaluated. The typical PtPdRhRuCu had higher HER activity than PtPdRhRuCu-1 MMNs with small mesopores and PtPdRhRuCu-2 MMNs with oversized mesopores (**Figure S26a**). The pore size influenced the specific surface area (**Figure S17**) and mass/electron transportation and played an important role in the density of the external exposure HEASs. As shown in **Figure S26b**, PtPdRhRuCu MMNs had a higher ECSA/EWSA (EWSA: electrochemical wettable surface area) ratio (ECSA and EWSA were proportional to the double layer capacitance values measured under dynamic and static conditions, respectively) than PtPdRhRuCu-1 and PtPdRhRuCu-2 MMNs, indicating the best ion migration in PtPdRhRuCu MMNs.⁴³ Additionally, the CV scan rate dependency of the capacitance indicated that the PtPdRhRuCu MMNs maintained higher capacitance levels even under severe dynamic conditions; this finding suggested the excellent ionic paths in PtPdRhRuCu MMNs (**Figure S26c**). Although the

PtPdRhRuCu-1 MMNs had higher surface areas than the PtPdRhRuCu MMNs (Figure S17d, e), the densities of exposed HEAS in the PtPdRhRuCu-1 MMNs were lower than those in the PtPdRhRuCu MMNs, probably due to the limited pore sizes of the PtPdRhRuCu-1 MMNs (Figure S27). The above results indicated that the appropriate mesopores in PtPdRhRuCu MMNs positively affected ion movement while ensuring high surface areas and the exposure of HEASs, improving HER performance.”

Comment 2-2: There may be a minor inconsistency in the atomic ratios. The ICP-OES and TEM-EDS results suggest that Ru takes the lowest atomic ratio of about 12-13%, while the SEM-EDS showed that Cu takes the lowest atomic ratio (Figure S7h).

Reply:

Sorry for our careless. The increased value should be Ru rather than Cu since that the Ru was the lowest be reduced. We already corrected this error, please see Figure S9 in the SI.

Comment 2-3: The atomic ratio obtained from ICP-OES was not consistently given, “23:22:20:13:22,” in line 149 and “21:23:21:13:22” in line 200.

Reply:

According to ICP-OES result (Table S1 in the SI), the ratio of Pt:Pd:Rh:Ru:Cu was 23:22:20:13:22. We have modified this point.

Comment 2-4: Compared to other elements, the XPS data of Pd appeared to be quite noisy. Is it because of the low amount of Pd near the surface? The authors might wish to add some more discussion.

Reply:

Yes, your observation is correct. Since Pd is first reduced and mainly present in the core of mesoporous nanospheres (Figure 3, S9), the amount of Pd near the surface is relatively small (Figure 8). We add more discussion in the MS [Page 9 Line 4–7] as: “Compared to other metals, the XPS signals of Pd appeared to be noisy due to the low content ratio of Pd near the surface. This phenomenon

occurred because Pd precursors were first reduced during the chemical reduction process to form a Pd-rich core, which will be further discussed later.”

Comment 2-5: Figure 2b, please detail how the mixed configuration entropy (ΔS_{mix}) for each point was calculated, and how the error bar was obtained.

Reply:

The atomic ratio for each element at a point of the selected area in **Figure 2b** in the **MS** and **Figure S27a2–c2** in the **SI** could be got as follow steps: (1) Open the free software ImageJ; (2) Open the specified file (.mrc) with ImageJ; (3) Select one point on the HAADF-STEM image and the value of each pixel is the atomic ratio for element in this point. The above details were presented in the **Note for Figure S27**. After the atomic ratio for each element was obtained, the ΔS_{mix} could be expressed as the following (*Nat. Rev. Mater.* **2019**, *4*, 515-534):

$$\Delta S_{\text{mix}} = -R \sum_{i=1}^n x_i \ln x_i$$

where R was the gas constant and x_i represented the molar concentration of each elemental component.

Select at least three points within the selected area to obtain error bar, which was supplied in the caption of **Figure 2** in the **MS** as: “Error bars in (a2–c2) based on the measurements at three points within the selected region”.

Comment 2-6: Some technical terms should be explained briefly so that general readers can understand, such as “ q ” in line 116 and “Löwdin charge” in line 204.

Reply:

Indeed, some scientific terms should be interpreted to apply to wider readers. As suggested, we explain the details of “ q ” in the caption of **Figure S4** in the **SI** as: “(The position of each pixel on the SAXS image was converted into the scattering angle 2θ or the scattering vector q (its modulus was $q = 4\pi \sin(\theta)/\lambda$, where λ is the wavelength of the X-ray). And q was more common used because the value of θ varies in a small range.)”

Löwdin charge was study the charge transfer effects. It depicted the total number of valence electrons for each element. By subtract it to the initial valence electrons of the atom would produce

the partial charge of each atom. Please see under **Table S7** in the SI and text in the **MS [Page 16 Line 4–7]** as: “In addition, the change in the valence electron density of each atom indicated electron transfer and redistribution upon the synergistic electronic coupling interactions in PtPdRhRuCu MMNs, as confirmed by Löwdin charge calculation, which was used to study the charge transfer effects (**Figure S25 and Table S7**).”

Comment 2-7: Table S3, please cite the source for the standard redox potentials of different metals (if applicable).

Reply:

Thanks for the advice, we cited the references (*J. Mater. Chem. A* **2015**, *3*, 18053-18058; *Chem. Eur. J.* **2016**, *22*, 7174-7178) for standard reduction potential of different half reactions for various metal precursors in **Table S2** in the **SI**.

Comment 2-8: Some figures were not mentioned or incorrectly cited in the main text. For instance, (1) Figure 2c was not cited in the text; (2) Line 155, “right site in Figure 1g”. Here, the authors may have wanted to refer to Figure 1f; (3) Line 192, “Figure 2a–e” should be revised into “Figure 3a–e. (4) Line 275, “Figure S13d” should be revised into “Figure S13f”; (5) Figure S9 and S10, the effects of temperature and the amount of HCl were not discussed in the main text; (6) Line 331, “Figure S14” was incorrectly referred to.

Reply:

Sorry for our careless. We have corrected one by one according to your carefully observation. The effect of temperature and amount of HCl were also discussed in the **MS [Page 11 Line 7–9]** as: “The content of Ru slightly increased with increasing reduction temperature, which had no effect on the mesoporous morphology (**Figure S11**). The appropriate amount of HCl (i.e., 6.0 M 0.4 mL) was selected in the present reduction system (**Figure S12**).”

Comment 2-9: Some minor writing issues (grammar and typos) need to be corrected. (1) Line 231, “71, 11%, and 16 at%” should be revised into “71, 11, and 16 at%”; (2) Line 257, “since the well

solubility of PEO and PMMA segments in DMF”; (3) Line 325, “astronger”; (4) Figure 5 caption, “chronoamperometry method” should be revised into “chronopotentiometry method”; (5) Figure S18c and S20a, y axis should be checked; (6) Line 411 and line 418, “absorbed” should be revised into “adsorbed”. (7) Line 417, “absorption” should be revised into “adsorption”.

Reply:

We appreciate your efforts in pointing out typos in the **MS**. We have carefully modified accordingly. Additionally, we have submitted the **MS** to the office of Nature Research Editing Service of Springer Nature Author Service to further improve grammar and spelling.

Comment 2-10: Simply out of curiosity, would it be suitable to call the reduction process “deposition”? The authors mentioned several times the terms such as deposition properties and deposition processes.

Reply:

Thank you for your valuable feedback on our manuscript. We agree that the terminology used in our original submission could be improved, and we appreciate the opportunity to address this issue.

Initially, we used the term "deposition" to describe the differences in reduction sequences of various metal elements, which we likened to a probably one-by-one deposition process. However, upon further reflection, we think that the term "reduction" is more appropriate for describing the nucleation and growth processes in our metal alloys.

Therefore, we have revised our manuscript and replaced instances of "deposition" with "reduction" where applicable in the **MS**. We believe that this change will enhance the clarity and accuracy of our research findings.

Thank you again for your helpful feedback, and we look forward to hearing your thoughts on the revised manuscript.

Reviewer #3:

Comment 3-1: Handily, the work compares many catalysts, their compositions and structures within the supplementary information under basic and acidic conditions and where possible more neutral pHs. One aspect of the work that wasn't discussed was how these HEAs under review were loaded onto the

carbon support for some measurements. Was this strategy successful in terms of reduced leaching or nanoparticle stability. Does the conductivity of the carbon influence the measurements?

Reply:

We agree with your comments on the electrochemical test details and believe that providing test details is rigorous and necessary. In electrocatalysis, the degree of dispersion of metal powders on an electrode, especially on glassy carbon electrodes, is important for obtaining reproducible catalytic activity. However, the stability of pure metal powders is difficult to maintain within the inks; spontaneous agglomeration tends to take place. This leads to an uncontrollable dispersion on the glassy carbon electrode (GCE). Therefore, the metal-based powders are usually mixed with carbon black (Vulcan XC-72) to improve stability and dispersion (*Nat. Synthesis* **2023**, 2, 119–128; *Nat. Catal.* **2019**, 2, 304–313), as well as probably enhance conductivity. The pure carbon black in our work shows a negligible catalytic performance for HER, which indicates that the metal powders the only mesoporous metal powders are acting as a HER catalyst.

In order to improve clarity, we explained the role of carbon black as: “Considering that the pure mesoporous metal nanospheres tended to agglomerate in the solution and were poorly dispersed on the GCE, the use of conductive Vulcan XC-72 carbon (without HER activity) ensured a uniform dispersion of electrocatalysis on the GCE.” Please see the **[Page 20 Line 6–9]** in the **MS**.

Comment 3-2: On reflection, the identification of atomic step/kink sites (Fig. S5) appears reasonable to assist in explaining the high activity. Whilst this may be a contributing factor, how confident are the authors that this is retained over the testing periods used? Rearrangement of surfaces as seen in Fig. S5 is not uncommon when catalysts are under testing conditions. The long-term testing data (Fig. S22) would suggest that this is potentially not the only factor due to the stability shown. However, Figure S21 does not have the same scope to address this. Local surface changes on the atomic scale may not be the strongest factor but the greater bulk or intermediary depth structure such as having a greater density of different metals about the pore structure do appear to matter as Figure S19 demonstrates.

Reply:

Thank you for your thoughtful observations and comments. Mesoporous materials, as a typical type of nanomaterial, exhibit high performance due to their unique pore effects, and their rough surfaces with abundant atomic steps/kinks may lead to lattice compression or high-index sites (*Adv. Sci.* **2015**, *2*, 1500112; *Nat. Protoc.* **2020**, *15*, 2980–3008), which could be a contributing factor to their catalytic activity. We supplemented HRTEM and SAXS data of PtPdRhRuCu after stability tests to investigate structural changes. As shown in **Figure S28b**, the PtPdRhRuCu MMNs after the long-term test still retains its atomic steps/kinks. The results of TEM, XRD, SAXS, ICP, and XPS confirm the stability of the composition and structure of PtPdRhRuCu MMNs. Please see data in **Figures S28, S29** in the **SI**.

One important reason for the improved stability of multimetallic metal alloys such as high-entropy alloys is the differences between the constituent atoms. Increasing disorder can likely suppress the migration of active centers and improve stability (*Science* **2022**, *376*, 151). Nevertheless, we acknowledge that surface restructuring of catalysts is a common phenomenon. Therefore, our concerns were also expressed in the **SI**, as show in the **Note for Figure S28** as: “The mesoporous morphology of PtPdRhRuCu MMNs were maintained after HER test. The atomic steps and kinks were still observed in **Figure S28b**, which might be a contributing factor for the stability. However, considering that local surface changes on the atomic scale on nanomaterials probably occurred during the electrochemical testing, rearrangement of surfaces of the electrocatalysts was not uncommon. The stable bulk morphology, composition, and porous structure of the nanosphere might be the main factor for maintaining long-term HER measurement.”

Comment 3-3: The manuscript offers a promising synthetic route to mesostructured HEAs and the activity achievable for HER under various conditions is valuable. I recommend acceptance following minor corrections: 1) 86 Until now, it is still a grand challenge to precise design of HEA...rewrite needed; 2) 166 resulting in an overlap each-other; 3) 194 dominant metallic states along with slightly oxidation states; 4) 249 but with a much low Ru content.

Reply:

Thank you for your positive feedback on our manuscript and for providing us with valuable suggestions for improvement.

We have carefully reviewed your comments and made the necessary revisions to the **MS**. Please see:

[Page 4 Line 2–4] “Nevertheless, the synthesis of MMA mesoporous nanospheres (MNs) with multiple elements under milder synthesis conditions, such as wet-chemical reduction in the solution phase, is still rarely reported.”

[Page 8 Line 4–6] “Unlike the previously reported single-crystal HEA (CrMnFeCoNi),²⁸ for which it was possible to obtain atomic resolution EDS maps, this process was more difficult in our MMN sample due to the presence of many different types of atoms overlapping each other.”

[Page 8 Line 27–28] “As shown in **Figure S8**, all the metal elements in the PtPdRhRuCu MMNs showed dominant metallic states.”

[Page 11 Line 4–9] “Different conditions, such as reducing agents, organic solvents, and acids, affected the final morphology (**Figure S10d–g**). For example, the strong reducing agent (dimethylamine borane) yielded agglomerated nanoparticles (**Figure S10d**), and formic acid generated an irregular porous structure (**Figure S10e**). The content of Ru slightly increased with increasing reduction temperature, which had no effect on the mesoporous morphology (**Figure S11**). The appropriate amount of HCl (i.e., 6.0 M 0.4 mL) was selected in the present reduction system (**Figure S12**).”

Reviewer #4:

Comment 4-1: There are several simple/grammatical errors which should be corrected during the revision process (e.g., absorbed vs adsorbed etc.).

Reply:

Sorry for our careless. We have corrected the typos/errors carefully. And we submitted the manuscript to the office of Nature Research Editing Service for “GOLD” level English editing.

Comment 4-2: The authors should avoid using terms like “impressively”, “remarkable” etc.

Reply:

We agree that using subjective terms such as “impressively”, “unique”, “novelty” and “remarkable”, etc., in academic writing may reduce the objectivity and scientific rigor of the article. It is a reasonable suggestion to avoid using such words, as it helps to make the article more objective and accurately describe the research results. We have modified and avoided the use of this type of word in the **MS** and **SI**.

Comment 4-3: The XPS analysis is less convincing. Some of the peak area ratios seem not to be correct (see e.g., Fig. 3d for Ru4+). This will directly affect the quantitative results.

Reply:

Thank you for your message regarding our research on X-ray photoelectron spectroscopy (XPS). We agree that XPS is a relatively effective technique for detecting surface information of a material. However, we acknowledge that XPS only samples a specific area during the testing process, which may affect the accuracy of the results. Additionally, spectrum fitting involves many factors that need to be considered, such as the full width at half maximum, peak ratio, fitting coefficients, and methods, which can introduce human factors that cannot be ignored.

We apologize for neglecting the ratio issue in the fitting results of Ru, i.e. the peak area ratio of Ru $3p_{3/2}:3p_{1/2}$ is 2:1, which has now been corrected. Please see the revised data in **Figure S8** in the **SI**. Furthermore, in the quantitative analysis of XPS, the entire peak area value of a certain element is often taken, such as Pt $4f$, Ru $3p$, and Cu $2p$, which is not directly related to the fitting data of a single element. We can confirm that the elements in the alloy mainly exist in the metallic state. Due to the complexity of the elements in our multimetallic alloys, we have decided to remove the XPS data from the **MS** but retain some of the text [**Page 8 Line 27–28**] as: “As shown in **Figure S8**, all the metal elements in the PtPdRhRuCu MMNs showed dominant metallic states.”

In addition, we also account for the reduced content of Pd in XPS considering the gradient distribution of the element in our mesoporous multimetallic alloys as: “Compared to other metals, the XPS signals of Pd appeared to be noisy due to the low content ratio of Pd near the surface. This phenomenon occurred because Pd precursors were first reduced during the chemical reduction process to form a Pd-rich core, which will be further discussed later.” Please see [**Page 9 Line 4–7**] in the **MS**.

Comment 4-4: Is there an optimum in pore size/BET surface area regarding the catalytic activity? What kind of pore structure is more advantageous in this context (regular vs irregular)?

Reply:

Thank you for your suggestion. Indeed, the pore size not only affects the specific surface area and the transport of reactants and products, but also influences the diffusion of ions and electrons. The pore size plays an important role in the catalytic reactions. Based on this, we have supplemented the SEM, TEM, BET characterization of three samples, including PtPdRhRuCu-1 (small mesopores), PtPdRhRuCu (typical middle mesopores), and PtPdRhRuCu-1 (large mesopores). The results showed that as the pore size increased, the specific surface area gradually decreased, which could lead to a decrease in active sites. However, the enlargement of the pore size is beneficial to mass transfer and exposure of high-entropy sites, just like the previously reported Pt catalyst with big pores, which is advantageous for the diffusion of gas products (*Nat. Commun.* **2013**, *4*, 2473). Considering multiple factors, we found that the typical PtPdRhRuCu exhibited the best HER activity.

Please see the data in **Figures S3, S7, S17, S26** in the **SI**, and the text in the **MS** as following:

[Page 12 Line 7–17] “The small molecular weight (5500) of PMMA decreased the average pore size to 8 nm, and it could be expanded to 41 nm when the large molecular weight (22000) of PMMA was used (**Figure S17a–c**). The PtPdRhRuCu samples with small and large pore sizes were denoted as PtPdRhRuCu-1 and PtPdRhRuCu-2 MMNs, respectively. The variations in the porous structures in PtPdRhRuCu-1, PtPdRhRuCu (typical), and PtPdRhRuCu-2 MMNs were observed in the N₂ adsorption–desorption isotherms and corresponding pore size distributions (**Figure S17d–f**), which were consistent with the HAADF–STEM results (**Figure 1e** and **S17g, h**). The Brunauer–Emmett–Teller (BET) surface areas of PtPdRhRuCu-1, PtPdRhRuCu (typical), and PtPdRhRuCu-2 MMNs were 39, 27, and 18 m² g⁻¹, respectively. The EDS maps in **Figure S17g, h** revealed that the element distributions of both PtPdRhRuCu-1 and PtPdRhRuCu-2 MMNs were similar to those of typical PtPdRhRuCu MMNs.”

[Page 16 Line 8–24] “The influences of the porous structures of MMNs (i.e., PtPdRhRuCu-1, PtPdRhRuCu, and PtPdRhRuCu-2 MMNs) on HER performance were evaluated. The typical

PtPdRhRuCu had higher HER activity than PtPdRhRuCu-1 MMNs with small mesopores and PtPdRhRuCu-2 MMNs with oversized mesopores (**Figure S26a**). The pore size influenced the specific surface area (**Figure S17**) and mass/electron transportation and played an important role in the density of the external exposure HEASs. As shown in **Figure S26b**, PtPdRhRuCu MMNs had a higher ECSA/EWSA (EWSA: electrochemical wettable surface area) ratio (ECSA and EWSA were proportional to the double layer capacitance values measured under dynamic and static conditions, respectively) than PtPdRhRuCu-1 and PtPdRhRuCu-2 MMNs, indicating the best ion migration in PtPdRhRuCu MMNs.⁴³ Additionally, the CV scan rate dependency of the capacitance indicated that the PtPdRhRuCu MMNs maintained higher capacitance levels even under severe dynamic conditions; this finding suggested the excellent ionic paths in PtPdRhRuCu MMNs (**Figure S26c**). Although the PtPdRhRuCu-1 MMNs had higher surface areas than the PtPdRhRuCu MMNs (**Figure S17d, e**), the densities of exposed HEASs in the PtPdRhRuCu-1 MMNs were lower than those in the PtPdRhRuCu MMNs, probably due to the limited pore sizes of the PtPdRhRuCu-1 MMNs (**Figure S27**). The above results indicated that the appropriate mesopores in PtPdRhRuCu MMNs positively affected ion movement while ensuring high surface areas and the exposure of HEASs, improving HER performance.”

Comment 4-5: Statistics should be reported for the electrochemical measurements.

Reply:

Yes, reporting statistics such as standard errors, can provide important information about the variability and reproducibility of the electrochemical measurements. So, we reported the statistics as: “Error bars obtained from three independent experiments.” Please see the caption in **Figure 4** in the **MS** and **Figure S30** in the **SI**.

Comment 4-6: Have the authors considered using a DOI approach to tailor the composition for maximum activity?

Reply:

We hope we understand your meaning correctly. "Using the DOI method to tailor the composition for maximum activity" refers to using the DOI method to identify which components have the greatest impact on producing a compound or catalyst with maximum catalytic activity and adjusting these components to achieve the best response.

In order to prove the advantage of multimetallic alloys, we compared the activity of our PtPdRhRuCu MMNs with other metal combinations to develop an experimental screening method. We started with a single metal material and found that metallic Pt was the best HER catalyst. Then, we gradually optimized the Pt-based alloy composition through the screening of binary, ternary, quaternary, and quinary alloys, similar to the process of alchemy that you mentioned. Eventually, we identified the advantages of optimal PtPdRhRuCu in comparison of other low-complexity alloys.

Please see details in **Figure S13–S16, S20, and Table S3, S4** in the **SI**, as well as the text in the **MS** as:

[Page 11 Line 12–24] “As mentioned above, in the preparation of porous metallic alloys by a wet-chemistry strategy based on micellar self-assembly, the complexity and difficulty of the reduction system increased with the addition of metallic elements. Finding suitable reducing conditions to balance the complex kinetic behaviors of different metals in various metal alloys was key to the synthesis of a well-defined mesoporous morphology. **Figures S13–S16** show the SEM images and XRD patterns of other monometallic (i.e., Pt, Pd, Rh, Ru, and Cu) and metallic alloys (i.e., bimetallic PtM (M = Pd, Rh, Ru, Cu), trimetallic PtRuM (M = Pd, Rh, Cu), and tetrametallic PtRhRuM (M = Pd, Cu)) prepared by fine tuning the chemical reduction system (see preparation details in Supplementary Information and **Tables S3 and S4**). Reaction conditions, such as the choice of precursor and polymer, type of acid, and reaction temperature, changed depending on the different composition of the Pt-based metallic alloy. The XRD results showed that the incorporation of Cu into the Pt lattice could be the main factor in the positive shift of the XRD peaks (**Figure S14e**), probably due to the smaller atomic radius of Cu than other noble metals (i.e., Pd, Rh, and Ru).”

[Page 14 Line 8–13] “We first evaluated the HER performance of monometallic catalysts and found that Pt MNs exhibited the highest activity among Rh MNs, Pd MNs, Ru NPs, and Cu NPs (**Figure S20a**). Then, Pt-based alloys (including binary, ternary, quaternary, and quinary alloys) were

screened step-by-step to determine the optimum HER catalyst in alkaline media (**Figure S20b–d**). As a result, PtPdRhRuCu MMNs showed the highest intrinsic HER activity and ion migration among the Pt-based catalysts (**Figure S20e, f**).”

Reviewer #5:

Comment 5-1: Starting from the title, the main claim is the formation of HEA mesoporous materials that is defined core-shell. This is misleading since the mesoporous particles are indeed multimetallic but with composition gradients from the outer to the inner part in which the conditions for HEA are probably met only in a small part of the material. This is even more problematic when discussing about catalytic activity (discussed later) For this reason, the authors should provide evidence that homogenous mesoporous HEA can be made. In alternative, the claim needs to be tone down. The title and the all discussion must be changed by removing the claim of HEA.

Reply:

Thank you for your feedback, it has been very valuable to us.

As suggested, we have revised the title to avoid directly using "High-Entropy Alloy (HEA)", and the title was changed from “Mesoporous High-Entropy Alloys” to “Mesoporous Multimetallic Nanospheres with Exposed High-Entropy Alloys Sites”.

Since the landmark research paper report about HEAs was published in 2004 by Ye et al., HEAs have attracted great attention both in academia and industry (*Adv. Eng. Mater.* **2004**, *5*, 299). The HEAs are alloys formed by five or more equal or approximately equal amounts of metals. The latest research result shows that the atomic concentration of each element in HEAs is 5 to 35% (*Prog. Mater. Sci.* **2021**, *120*, 100754) and the composition has been expanded to include other complexes such as metal oxides (*Science* **2022**, *378*, 1320–1324), core-shell structure (*Adv. Funct. Mater.* **2022**, 2204643; *ACS Nano* **2020**, *14*, 15131–15143), phosphides, and even high-entropy single atoms (*Nat. Commun.* **2022**, *13*, 5071). More recently, the high-entropy electrolyte has also been published by Yang et al. (*Nat. Sustain.* **2023**, *6*, 325–335).

Nevertheless, we acknowledge that the current material may not have a homogeneous HEA composition and further investigation is needed to determine the extent to which the conditions for

HEA formation are met. In fact, HEA is not defined by a requirement for a homogeneous and equimolar ratio composition, and its structure can be either crystalline or amorphous. Although our PtPdRhRuCu may belong to a class of alloys that could be considered as HEA based on ICP and XRD results, we agree that a multimetallic alloys may be a better description due to the composition gradients. However, we cannot ignore the role of high-entropy alloys sites (HEAS) of PtPdRhRuCu in catalysis. Our results demonstrate that even though HEAS may be concentrated in certain regions of the pores, they still exhibit highly effective catalytic activity. Therefore, we have replaced the term HEA with a multimetallic alloys that contains HEAS throughout the manuscript.

Comment 5-2: In general, the materials are not homogenous and the description of structure/compositions of the materials is confusing. For instance, it is still unclear if the porous walls are polycrystalline (as mentioned in page 5 line 117) or if the full porous particle is made of a unique porous monocrystal (figure 2c).

Reply:

Thank you for your comment. We agree that our materials are not homogeneously distributed, which makes discussing their composition and structure more complex and challenging. We are aware that there are various types of alloys, including homogeneous alloys, random alloys, intermetallic alloys, and single-atom alloys, etc. (*Adv. Sci.* **2022**, *9*, 2104054). When it comes to core-shell alloys, there are types such as phase-separated core-shell alloys, dynamically evolving alloys, heterogeneous phase alloys, and so on (*Acc. Chem. Res.* **2020**, *53*, 2913–2924). Although our PtPdRhRuCu alloy has a composition gradient, no phase separation occurs (**Figure 1c, h**). To illustrate this process clearly, we have added Pt-based binary, ternary, and quaternary alloys and analyzed the evolution of crystal phases by XRD (**Figure S13–S16**). The results show that PtPdCu is the main element that constitutes the *fcc* polycrystalline structure, while Rh and Ru may randomly scatter around the main element without affecting the crystal phase. Certainly, to better represent our material, as you suggested in your previous comment, we used a multimetallic alloys instead of HEA in the whole **MS**.

Please see **Figure 1** in the **MS**, **Figure S13–S16** in the **SI**, and the details as: “As mentioned above, in the preparation of porous metallic alloys by a wet-chemistry strategy based on micellar self-

assembly, the complexity and difficulty of the reduction system increased with the addition of metallic elements. Finding suitable reducing conditions to balance the complex kinetic behaviors of different metals in various metal alloys was key to the synthesis of a well-defined mesoporous morphology. **Figures S13–S16** show the SEM images and XRD patterns of other monometallic (i.e., Pt, Pd, Rh, Ru, and Cu) and metallic alloys (i.e., bimetallic PtM (M = Pd, Rh, Ru, Cu), trimetallic PtRuM (M = Pd, Rh, Cu), and tetrametallic PtRhRuM (M = Pd, Cu)) prepared by fine tuning the chemical reduction system (see preparation details in Supplementary Information and **Tables S3 and S4**). Reaction conditions, such as the choice of precursor and polymer, type of acid, and reaction temperature, changed depending on the different composition of the Pt-based metallic alloy. The XRD results showed that the incorporation of Cu into the Pt lattice could be the main factor in the positive shift of the XRD peaks (**Figure S14e**), probably due to the smaller atomic radius of Cu than other noble metals (i.e., Pd, Rh, and Ru).” in the **[Page 11 Line 12–24]** in the **MS**.

We think that the porous walls are also polycrystalline and not phase separated, as confirmed by the FFT result inset of **Figure 1d**. The region selected **Figure 1d** probably show the outside porous region. The numerous mesoporous walls and channels formed the final mesoporous nanospheres. The SAED pattern of multiple nanospheres inset of **Figure 1c** also confirmed the polycrystalline of PtPdRhRuCu, which consistent with XRD result in **Figure 1h**. Please see details as following:

[Page 5 Line 9–10] “The corresponding fast Fourier transform (FFT) patterns (selected from the outside porous wall region) (inset **Figure 1d**) demonstrated the *fcc* structures of PtPdRhRuCu MMNs.”

[Page 7 Line 1–3] “The powder X-ray diffraction (XRD) pattern (**Figure 1h**) shows that the PtPdRhRuCu MMNs exhibited metallic *fcc* structures with four diffraction peaks at $2\theta = 40.7, 47.2, 69.3,$ and 83.2° that were assigned to the (111), (200), (220), and (311) planes, respectively, which consistent with TEM result.”

Comment 5-3: On the novelty: polymer templated HEA mesoporous materials with homogenous composition have been recently reported in article entitled High-Entropy-Alloy Nanocrystal Based Macro- and Mesoporous Materials ACS Nano 2022, 16, 10, 15837–15849 referenced by the author in their manuscript (ref 31). The justifications used to "undermine" this recent work (not compatible with

Cu and particle size due to annealing) are inaccurate. The approach proposed by Kang and co-workers still present some merits as respect to the previous work in term of synthetic conditions (in solution, milder conditions) and this can be highlighted in the introduction. However, the very same approach was reported by some of the authors in many articles for fabrication of mesoporous particles made of metals (mesoporous Rh Nat Comm 2017, mesoporous Pd Chem. Sci.2019, 10, 4054), and several combinations of alloys (PdCu ACS Appl. Mater. Interfaces 2019, 11, 40, 36544–36552, RhCu Chem. Mater. 2018, 30, 2, 428–435, PtCu <https://doi.org/10.1002/chem.201804305> or even Pt Pd Rh <https://doi.org/10.1246/bcsj.20190316>). This work is a straightforward implementation of the same concept for multi metallic materials (and this can be ok) but with a lack of control and homogeneity.

Reply:

We agree that the statement made to undermine the recent work on polymer templated HEA mesoporous materials (*ACS Nano* **2022**, *16*, 15837–15849) is not accurate. We acknowledge the merits of the approach proposed by our strategy in terms of synthetic conditions, such as the use of milder conditions in the solution. Our method also offers advantages in terms of control the particle and pore size of the multimetallic alloys. The text in the **MS** was revised as [**Page 3 Line 30 and Page 4 Line 1–4**]: “Very recently, a mesoporous noble metal-based MMA (PtPdRhRuIr HEA) with a uniform pore size and a large surface area was successfully synthesized using polymer-templated spray-drying through following the annealing strategy by Faustini *et al.*²⁵ Nevertheless, the synthesis of MMA mesoporous nanospheres (MNs) with multiple elements under milder synthesis conditions, such as wet-chemical reduction in the solution phase, is still rarely reported.”

While the polymer templated method (*ACS Nano* **2022**, *16*, 15837–15849) can produce mesoporous/macroporous HEA materials, their approach relies on annealing process, leading to a non-uniform particle size distribution ranging from 500 nm to 5µm. In contrast, our wet-chemical reduction method is gentler and simpler, and produces more uniform nanoparticles (~128 nm, as shown in **Figure S3** in the **SI**), which has been rarely reported.

In addition, although different methods may utilize similar mechanisms (such as, sol-gel strategy for porous SiO₂, alcohol heat method for metallene), there are often significant differences in the details of the preparation process. While we employed a similar surfactant self-assembly approach as

we previously reported to prepare the multimetallic alloys, there are still significant differences between the preparation of different alloys. For instance, as you mentioned the preparation of mesoporous Pd, the copolymer types, precursor types, and temperatures, etc. all play important roles in the synthesis process, and these factors can vary significantly even between similar methods used to prepare different alloys. The nucleation and growth of metals when the kinds of precursors are increased and complexity is very different from low entropy alloys such as binary and ternary, yet it is rarely reported. To demonstrate the differences more clearly in the preparation conditions for different metals or alloys, we have included a detailed list of reaction parameters for each material in **Table S4** in the **SI**.

We acknowledge that achieving uniformity in multi-component materials synthesis can be a challenging task, particularly in balancing differences in reduction kinetics between different metals. For instance, it has been recently published an article reporting the control of Ru–Ir–Pt reduction kinetics for achieving uniformity (*J. Am. Chem. Soc.* **2022**, *144*, 4224–4232). When the elements increased to five or more, controlling the crystal structures of solid-solution alloys with the same composition is a highly challenging task.

Moreover, in catalysis, it is crucial to consider the interface interaction between the reactants/intermediates and the catalyst, rather than just the catalyst's surface. Despite in some non-porous catalysts, the core-shell composition can significantly affect catalytic performance (*J. Am. Chem. Soc.* **2021**, *143*, 11262–11270; *Nat. Mater.* **2007**, *6*, 692.) maybe due to the core-shell interface. Our exposed mesoporous structure provides numerous interface contact sites for reactant and therefore the high entropy sites are more accessible.

In addition, although there may be a gradient distribution of elements within individual porous spheres, we characterized multiple metal nanospheres and found that they possess similar morphology and element distribution patterns, as confirmed in **Figure S7** in the **SI**. This observation suggests that the distribution in individual spheres can be extended to other porous spheres, and the structure remains stable over a long reaction time (**Figure S28, S29** in the **SI**).

Overall, we acknowledge that the novelty of this research may not be groundbreaking. However, we have still presented some novel aspects of combining mesoporous structure and compositional

effects, including: (1) our innovative use of soft template-directed self-assembly to prepare mesoporous multimetallic alloys, which, to our knowledge, is likely the first report of such a method using a chemical reduction approach; (2) our detailed discussion of the nucleation and growth characteristics of multimetallic alloys during the reduction process (**Figure 3** and **Figure S9**), which is a crucial aspect that cannot be predicted solely based on known data and must be analyzed in the context of the specific problem at hand; (3) our exploration of the impact of composition and pore structure on HER performance, which demonstrates the superiority of multimetallic alloys over other mesoporous metals, including mono-, bi-, tri-, and tetra-metallic systems, due to their complexity and synergistic effects (**Figures S13-S17, S26, S27** in the **SI**). This process can help readers better understand the cocktail effect of multiple elements, which has been rarely reported in previous studies of nanostructured multimetallic alloys (e.g., *Adv. Mater.* **2023**, *35*, 2209242; *J. Am. Chem. Soc.* **2022**, *144*, 11525–11529; *J. Am. Chem. Soc.* **2022**, *144*, 10582–10590; *ACS Nano* **2022**, *16*, 15837–15849; *Nat. Commun.* **2021**, *12*, 6261).

We appreciate your valuable insights, and we understand that there are still some shortcomings and challenges that need to be further explored.

Comment 5-4: The "deposition" mechanism need to be clarified. It is shown that Ru is the slowest to be deposited. In this case the author should discuss how Ru can be inserted in the fcc lattice together with the other metals (considering that Ru that crystallize in different crystal lattice).

Reply:

It's really a question worth thinking about. The deposition mechanism, particularly the slower reduction rate of Rh and Ru compared to Pd, Pt and Cu, needs further clarification. Considering that Ru crystallizes in a different crystal lattice, it is important to discuss how it can be inserted into the *fcc* lattice along with the other metals.

Our experimental results (**Figure 3** and **Figure S9**) show that the reduction order followed: Pd > Pt \approx Cu > Rh > Ru. The slower deposition rate of Ru can be attributed to the lower standard reduction potential than other noble metals (i.e., Pd, Pt, and Rh), except for the special case of Cu. In fact Ru precursors are also gradually consumed during the reduction process, while Ru ions are reduced more

when the relative concentration of other metals is low, thus forming the shell structure. It should be noted that both Rh and Ru precursors are gradually reduced during the entire reduction process, but when the concentrations of Pd, Pt, and Cu precursors are higher, they are preferentially consumed, and more Rh and Ru are reduced when the concentrations of other metal precursors are lower, thus mainly enriching in the shell.

XRD results in **Figures S14-S16** confirmed that the *fcc* lattice parameters of PtPdRhRuCu are mainly determined by PtPdCu, and the addition of Rh and Ru does not cause any significant change. Similarly, in PtRh and PtRu alloys, the lattice parameter of Pt remains unchanged upon the addition of Rh and Ru. This may be due to the fact that the atomic radii of Ru and Rh are close to that of PtPdCu, or because Ru and Rh are doped into the interstices of PtPdCu.

Please see the revised text in the **MS** as:

[Page 10 Line 17–18] “As Pd, Pt and Cu were preferentially consumed, Rh and Ru gradually grew more on the outside of the nanosphere than on the inside without changing the lattice parameters.”

[Page 10 Line 21–24] “It should be noted that Rh and Ru precursors were gradually reduced throughout the process, but were more consumed at lower concentrations of Pd, Pt, and Cu precursors, resulting in the enrichment of Rh and Ru in the shell.”

[Page 11 Line 16–24] “**Figures S13–S16** show the SEM images and XRD patterns of other monometallic (i.e., Pt, Pd, Rh, Ru, and Cu) and metallic alloys (i.e., bimetallic PtM (M = Pd, Rh, Ru, Cu), trimetallic PtRuM (M = Pd, Rh, Cu), and tetrametallic PtRhRuM (M = Pd, Cu)) prepared by fine tuning the chemical reduction system (see preparation details in Supplementary Information and **Tables S3 and S4**). Reaction conditions, such as the choice of precursor and polymer, type of acid, and reaction temperature, changed depending on the different composition of the Pt-based metallic alloy. The XRD results showed that the incorporation of Cu into the Pt lattice could be the main factor in the positive shift of the XRD peaks (**Figure S14e**), probably due to the smaller atomic radius of Cu than other noble metals (i.e., Pd, Rh, and Ru).”

Please also see the **Note for Figure S14** in the **SI**: “Compared to Pt (PDF#04-0802), the XRD peaks of PtPd do not significantly shift probably due to a low lattice mismatch of 0.77%⁸. However, it is interesting to note that we did not find a shift in the XRD peaks of PtRu and PtRh except for PtCu,

which might be explained by the fact that Pt dominates in the alloys and the atomic radii of Ru and Rh are only slightly smaller than Pt (atomic radii order: Pt (130 pm) > Pd (128 pm) > Rh/Ru (125 pm) > Cu (117 pm)⁹. Although the XRD peaks of PtRh and PtRu are not shifted, their full width at half maximum (FWHM) become larger, compared to PtPd alloy. According to Sherrer's formula, a larger FWHM represents a decrease in crystallite size¹⁰. The possible reason is the increase of defects caused in the introduction of Ru and Rh.”

Comment 5-5: In page 12 line 261 it is indicated that the metallic ion are already complexed by the PEO chain increasing the concentration around the micelle. If the precursors are already close to the micelles, this is in contradiction with the deposition mechanism shown in figure 4. In addition scheme in figure 4 c illustrate the formation of a single nanoparticle not of a porous material. The scheme can be modified to improve the clarity for the reader.

Reply:

Indeed, the PEO groups can locally increase the concentration of metal ions due to hydrogen bonding with water-soluble metal ions. Nevertheless, the reduction in multicomponent alloys still follows a certain order. Taking Pd and Rh as an example, when their concentrations are comparable, Pd will be preferentially reduced, leaving only a small amount of Rh reduced. As Pd is consumed, the relative concentration of Rh precursor sharply increases, resulting in more Rh being reduced.

The reduction process in multimetallic alloys is more complex, making it difficult to accurately describe the formation of porous alloys using a cartoon scheme. However, as suggested, we still try to modify the mechanism scheme to illustrate the reduction process and formation of porous structure more clearly. Please see **Figure 3c** in the **MS**.

Comment 5-6: On the electrochemical activity. DFT calculations have been performed to understand the high activity in alkaline media. However, the used model surface is based on a composition (HEA) that do not reflect the real surface of the catalyst that is not homogenous in composition. The multimetallic particles are indeed very active but the global activity is the results of the contribution of an infinite of combination of sites (gradient in composition) that can't be oversimplified as in figure

6. Here again, without providing mesoporous HEA with homogenous composition this discussion is not accurate and should be removed.

Reply:

Thank you for your careful evaluation of the DFT section. We agree with your comment that DFT calculations are typically based on classical models of single-crystal metals and their application to complex multi-metal systems is extremely challenging and limited. Theoretical calculations can often be difficult to simulate the actual structure, interface, and reaction environment. Despite the HEA sites are accessible in our PtPdRhRuCu due to its large exposed mesopores, it is not rigorous to use only the compositions of the HEA fraction to represent the whole catalyst. Therefore, we have decided to remove the DFT calculation results from the main text and instead provide a brief discussion.

Furthermore, we have provided additional explanations in the **SI** to clarify this issue as follows: "It should be noted that although we used the most commonly proposed mechanism with a well-defined and accepted model, the computational model become more difficult to build in complex multimetallic alloys, and the inclusion of additional factors in the model may influence the computational results."

Comment 5-7: The method was extended to other metals (Ni, Co, Mo) but the here again the term HEA can't be used without providing evidences (chemical mapping at high resolution, XPS, XRD...)

Reply:

Thank you for your comment. We agree that the term HEA (based on your previous comment, the multimetallic alloys was used instead of HEA in our cases) should only be used when sufficient evidence is provided to support the claim. In addition to the SEM results mentioned earlier, we have also included HAADF-STEM, EDS maps, XRD, and SAXS analyses for other multimetallic alloys, including PtPdCuNiCo and PtPdCuNiCoMo. These additional analyses provide further evidence to support our conclusions about the composition and structure of the materials as **[Page 12 Line 18–23]** in the **MS**: "Our synthetic strategy could be used to synthesize other MMNs, such as PtPdCuNiCo (**Figure S18**) and PtPdCuNiCoMo (**Figure S19**). Both the PtPdCuNiCo and PtPdCuNiCoMo MMN samples had nanospherical morphologies with uniformly exposed mesopores, as revealed by SEM, HAADF–STEM, and SAXS results. The XRD patterns in **Figure S18e and S19e** confirmed the single-

phase alloy structure without metal or metal oxide phase segregation in the PtPdCuNiCo and PtPdCuNiCoMo MMNs.” Please also see **Figure S18, S19** in the **SI**.

Comment 5-8: In addition the following typos can be corrected: (1) page 9 line 192 Figure 3; (2) page 4 line 93 tunable; (3) page 19 line448 tunable

Reply:

We apologize for our carelessness and have carefully revised the **MS**. Furthermore, we have submitted the **MS** to the English editorial office (“GOLD” level) for further editing to ensure readability.

Reviewer comments, second round -

Reviewer #1 (Remarks to the Author):

The authors have addressed most of my comments and questions in a satisfying way. I'm especially happy to see that they agree to remove the unrealistic DFT calculations. I also appreciate that they explain now their step by step approach towards increasing complexity of the synthesized structures. Therefore I can now recommend the manuscript for publication.

Reviewer #2 (Remarks to the Author):

I appreciate the authors' efforts in addressing the comments and improving the quality of the manuscript. I am happy with the changes and would like to support publication at Nat. Commun.

Reviewer #3 (Remarks to the Author):

Taking into account the comments the authors have provided and the corresponding changes, I can recommend publication.

Reviewer #4 (Remarks to the Author):

I am satisfied with the changes made and the additional information provided by the authors. The paper in its present form can be accepted for publication in Nature Communications.

Reviewer #5 (Remarks to the Author):

This is a review for the revised version of the manuscript now entitled " Mesoporous Multimetallic Nanospheres with Exposed High-Entropy Alloys Sites"

After carefully reading the revised version of the manuscript and the reply to the reviewer's comments, I can see that the manuscript has been improved a lot. Some of the main claims (mesoporous high entropy alloy) have been corrected. The authors made their best to improve this manuscript and clarify some technical aspects. This huge amount of work certainly deserves to be published. Is this work meeting the requirements of significance and novelty in the field to be recommended for Nature Communications? The core of the article is the synthesis of the multi metallic spheres. I understand the authors point of view, synthetic conditions are more challenging when combining 5 metals as respect to mono-, bi- or tri-metallic materials. However, as stated in the previous round, I still believe that the article lacks from novelty as respect to the established literature (mostly from the same group as listed in my previous comments) since the approach and the materials are very similar. This my personal opinion based on what I expect as a reader to find in Nat Comm but of course it is up to the editor to decide on the content.

Dear Reviewers,

Thank you for taking the time to review our manuscript again and providing your feedback. In the previous round, your valuable comments and suggestions helped us to improve the manuscript a lot. In addition, we have carefully checked the Manuscript (MS) and Supplementary Information (SI) to improve the quality again.

Sincerely,

Yusuke Yamauchi (*Highly Cited Researcher in Materials and Chemistry*)

Professor / The University of Queensland, Australia

https://scholar.google.com.au/citations?hl=en&user=568w7h8AAAAJ&view_op=list_works

Associate Editor / J. Mater. Chem. A, RSC

Co-Editors / Chem. Eng. J. Elsevier

Honorary Group Leader / National Institute for Materials Science (NIMS), Japan

Reviewer #1: The authors have addressed most of my comments and questions in a satisfying way. I'm especially happy to see that they agree to remove the unrealistic DFT calculations. I also appreciate that they explain now their step by step approach towards increasing complexity of the synthesized structures. Therefore I can now recommend the manuscript for publication.

Reviewer #2: I appreciate the authors' efforts in addressing the comments and improving the quality of the manuscript. I am happy with the changes and would like to support publication at Nat. Commun.

Reviewer #3: Taking into account the comments the authors have provided and the corresponding changes, I can recommend publication.

Reviewer #4: I am satisfied with the changes made and the additional information provided by the authors. The paper in its present form can be accepted for publication in Nature Communications.

Reply to #1~#4:

Thank you for your valuable comments and feedbacks in the previous round. It is our common goal to improve the quality of the manuscript, and we are happy that our revisions meet the requirements of you and the journal. Of course, during the revision process, we also learned a lot according to your comments, which will be beneficial for our further research in the future.

Reviewer #5: This is a review for the revised version of the manuscript now entitled " Mesoporous Multimetallic Nanospheres with Exposed High-Entropy Alloys Sites" After carefully reading the revised version of the manuscript and the reply to the reviewer's comments, I can see that the manuscript has been improved a lot. Some of the main claims (mesoporous high entropy alloy) have been corrected. The authors made their best to improve this manuscript and clarify some technical aspects. This huge amount of work certainly deserves to be published. Is this work meeting the requirements of significance and novelty in the field to be recommended for Nature Communications? The core of the article is the synthesis of the multi metallic spheres. I understand the authors point of view, synthetic conditions are more challenging when combining 5 metals as respect to mono-, bi- or tri-metallic materials. However, as stated in the previous round, I still believe that the article lacks from novelty as respect to the established literature (mostly from the same group as listed in my previous

comments) since the approach and the materials are very similar. This my personal opinion based on what I expect as a reader to find in Nat Comm but of course it is up to the editor to decide on the content.

Reply:

We appreciate your acknowledgement of our revisions and that you think our manuscript is accepted for publication in *Nature Communications*. The quality of the manuscript cannot be improved without the comments and suggestions from you and other reviewers. We have carefully thought about the main claims you motioned in the previous round, that is mesoporous high entropy alloy, and corrected it in our manuscript. Indeed, we understand that no research result is totally perfect. It has always been our goal to do our best to enhance the novelty and significance of our research results.

Again, we thank you for your comments and suggestions for improve our work.